# Condensin-mediated restriction of retrotransposable elements facilitates brain development in *Drosophila melanogaster*

Bert I. Crawford [1,3], Mary Jo Talley[1,3], Joshua Russman [1], James Riddle[1], Sabrina Torres[1], Troy Williams[1] & Michelle S. Longworth [1,2] ✉

Neural stem and progenitor cell (NSPC) maintenance is essential for ensuring that organisms are born with proper brain volumes and head sizes. Microcephaly is a disorder in which babies are born with significantly smaller head sizes and cortical volumes. Mutations in subunits of the DNA organizing complex condensin have been identified in microcephaly patients. However, the molecular mechanisms by which condensin insufficiency causes microcephaly remain elusive. We previously identified conserved roles for condensins in repression of retrotransposable elements (RTEs). Here, we show that condensin subunit knockdown in NSPCs of the *Drosophila* larval central brain increases RTE expression and mobility which causes cell death, and significantly decreases adult head sizes and brain volumes. These findings suggest that unrestricted RTE expression and activity may lead to improper brain development in condensin insufficient organisms, and lay the foundation for future exploration of causative roles for RTEs in other microcephaly models.

Brain development in both the fruit fly, *Drosophila melanogaster*, and mammals relies on the ability of neural stem and precursor cells (NSPCs) to asymmetrically divide (reviewed in refs. 1,2). This asymmetric division allows for self-renewal and the generation of additional cells which will differentiate into neurons or glia, thus allowing for a small number of stem cells to give rise to hundreds of cells that populate the adult brain and increase brain volume. Abnormal brain development, accompanied by changes in brain volume and head size, is a common characteristic of several neurodevelopmental disorders, including autism spectrum disorder (ASD), attention deficit hyperactivity disorder (ADHD), Rett syndrome, and microcephaly[3–11]. However, the underlying mechanisms responsible for the change in brain volume are not well understood.

Biallelic mutations and deletions of genes encoding subunits of the condensin complex I and condensin complex II have been identified in patients with microcephaly[12–16]. Furthermore, conditional, brain-specific knockout[17] or mutation[12] of condensin subunits caused reduced cortical volume and microcephaly in mice. While these studies revealed increased DNA damage and cell death in condensin-deficient neural progenitor cells, the exact causes of these phenotypes remain unclear. Condensins interact with DNA to promote efficient chromatin condensation and ensure that equal division of genetic material occurs during mitosis[18–23]. Our lab previously identified roles for condensin I and condensin II in the repression of retrotransposable elements (RTEs) in both human cells[24,25] and *Drosophila* cells and tissues[26]. RTEs are DNA elements that can copy and paste themselves into other areas of the genome through the process of retrotransposition which involves the generation of cDNA by an RTE-encoded reverse transcriptase[27]. RTEs are present in most organisms, and they make up approximately 40% of the human genome[28] and 30% of the *Drosophila* genome[29,30]. Interestingly, increased RTE expression and/or activity has been observed in tissues of patients with several

[1]Department of Inflammation and Immunity, Lerner Research Institute, Cleveland Clinic, Cleveland, OH 44195, USA. [2]Cleveland Clinic Lerner College of Medicine, Case Western Reserve University School of Medicine, Cleveland, OH 44195, USA. [3]These authors contributed equally: Bert I. Crawford, Mary Jo Talley. ✉e-mail: longwom@ccf.org

different neurodevelopmental disorders. Rett syndrome patients harbor mutations in *MECP2*, which contributes to DNA methylation of Long Interspersed Element 1 (L1) RTE copies; L1 copies were increased in postmortem brain tissue from patients with Rett syndrome[31,32]. L1 expression was also increased in autism spectrum disorder (ASD) patient tissue[33], and endogenous retrovirus expression was increased in attention deficit with hyperactivity disorder (ADHD) and ASD brains[34,35]. However, increased RTE expression and activity have not been investigated as a potential causative mechanism for the development of abnormal brain volumes associated with neurodevelopmental disorders.

In this study, we used a newly developed *Drosophila* model of microcephaly to investigate whether the unrestricted expression and activity of RTEs caused by condensin subunit insufficiency is responsible for the observed decreases in adult brain volume and head size. Our data show that significant loss of brain volume is first observed in condensin-insufficient pupae, and that condensin knockdown in NSPCs in the third instar larval central brain is necessary for the observed phenotypes. Excitingly, we show that RTE expression and mobility are increased in NSPCs in the third instar larval central brain. Like the previous studies in mammals[17], our results also demonstrate increased cell death in these NSPCs. Interestingly, increased levels of retrotransposition and increased cell death are rescued when condensin-insufficient larvae are allowed to develop on food containing nucleoside reverse transcriptase inhibitors (NRTIs) which inhibit RTE activity[36,37]. Finally, we show that condensin-insufficient larvae which are fed NRTIs can develop into adults with head sizes that are not significantly different from controls. These findings suggest that unrestricted RTE expression and mobility in condensin-insufficient NSPCs of the developing *Drosophila* central brain can lead to high levels of cell death, resulting in significant loss of brain volume and the development of microcephaly. Our studies, therefore, identify RTE-mediated cell death as a potential cause of microcephaly in *Drosophila*, and lay the foundation for investigation of this mechanism in human patient tissue and mammalian models of neurodevelopmental disorders.

## Results

### Condensin knockdown results in microcephaly in *Drosophila*
Similar to humans with condensin mutations, mutation or conditional knockout of condensin proteins in the developing murine brain causes microcephaly[12,17]. To determine whether the knockdown of condensin proteins results in microcephaly in *Drosophila melanogaster*, the condensin II protein, Cap-d3, was targeted by expressing dsRNAs in the developing brain. Ubiquitous knockdown of Cap-d3 in the fly is lethal in the first instar larval stage, and antibodies that reliably work to detect Cap-d3 protein in immunofluorescence analyses performed on *Drosophila* tissue are unavailable. Therefore, to confirm that the dsRNA was effective in knocking down *cap-d3*, qRT-PCR analyses of *cap-d3* transcript levels were performed in first instar larvae expressing *cap-d3 dsRNA* under the control of *armGAL4*, which drives expression in a ubiquitous manner during this early developmental stage. Indeed, results showed significant decreases in *cap-d3* transcripts in *cap-d3 dsRNA* expressing larvae, as compared to control, *GFP dsRNA* expressing larvae (Fig. 1a). To drive expression of *cap-d3 dsRNA* in the developing brain, a combined *eyGAL4, GMRGAL4* driver obtained from the Johnston lab at Johns Hopkins University was used. Both the Eyeless and GMR proteins are expressed in the developing brain[38–42]. Results demonstrated that adult female flies expressing *cap-d3 dsRNA* under the control of *eyGAL4, GMRGAL4* exhibited visibly smaller heads in comparison to control *GFP dsRNA* expressing flies (Fig. 1b). Female flies are shown, since males expressing *cap-d3 dsRNA* died at the mid to late pupal stages. As a proxy for measuring adult head size, the distance between two macrochaetes positioned between the eyes of adult flies was measured (pictured in Fig. 1c). Results demonstrated that Cap-

d3 insufficient, female, adult flies reared at either 25 °C (Fig. 1d) or 18 °C (Fig. 1e) exhibited significantly decreased head sizes. Measurement of dissected adult brain volumes from flies 1–3 days post-eclosure, revealed that Cap-d3 insufficient flies also possessed significantly smaller adult brains, as compared to controls (Fig. 1f).

To determine whether the knockdown of Cap-d3 was directly responsible for the observed decreases in head size and brain volume, genetic rescue experiments were conducted. A *UAS-GFP-cap-d3* allele was expressed in the background of *cap-d3* insufficiency to bring Cap-d3 expression levels closer to wild-type levels during development (Fig. 1g). Results demonstrated that female flies expressing *cap-d3 dsRNA* in combination with GFP-Cap-d3 (blue) exhibited head sizes that were not significantly different from control flies (black, pink, and turquoise). They were, however, significantly larger than the head sizes of flies expressing *cap-d3 dsRNA* alone (dark purple) or in combination with GFP protein (light purple).

*Drosophila* condensin I and condensin II both contain structural maintenance of chromosome proteins (Smc) Smc2 and Smc4, but differ in the chromosome-associated protein (Cap) subunits, with Cap-h, Cap-d2, and Cap-g comprising the Cap subunits of condensin I and Cap-h2 and Cap-d3 comprising the Cap subunits of condensin II (Fig. 2a). To determine whether decreasing the expression of other condensin subunits also results in microcephaly, head sizes of flies expressing dsRNAs targeting *smc2*, *cap-d2*, and *cap-h* were analyzed (Figs. 2b, c, e, g, i, k). The knockdown of the respective transcripts in first instar larvae were analyzed by qRT-PCR (Figs. 2d, f, h, j, l). Results confirmed that all dsRNAs efficiently targeted condensin subunits and demonstrated that decreasing each of the subunits also resulted in smaller female, pupal head sizes (knockdown was pupal lethal) (2c, g) or smaller female, adult head sizes (2e, i, k). Together, these results suggest that condensin I and condensin II proteins are necessary for proper brain development in *Drosophila*.

### Cap-d3 insufficiency leads to smaller pupal brain volumes
Given that *smc2* and *cap-d2* knockdown resulted in visibly smaller pupal heads, *cap-d3 dsRNA* expressing pupae were also examined. Comparisons between *cap-d3 dsRNA* expressing male pupae and controls (pupae were removed from their cases at 72–96 h post pupation) also revealed visibly smaller head sizes in Cap-d3 insufficient pupae (Fig. 3a). To determine whether larval brain volumes were also decreased by Cap-d3 insufficiency, the volumes of both female and male third instar larval brains expressing *cap-d3 dsRNA* were measured and compared to controls (Fig. 3b). Interestingly, no differences in brain volumes were observed for either sex, when compared to controls. These data suggest that microcephaly in condensin-insufficient brains first becomes apparent in the pupal stages, and the causative events most likely occur in the late larval stages.

To identify the cell types in which *eyGAL4, GMRGAL4* drives Cap-d3 insufficiency, subsequently leading to loss of brain volume and head size, immunofluorescence analyses were used to image the *eyGAL4, GMRGAL4*-driven expression pattern of GFP protein in third instar larval brains. Results demonstrated that the *eyGAL4, GMRGAL4* drives expression in a large region of the third instar larval central brain as well as parts of the optic lobes (Fig. 4a). Since this region includes deadpan (Dpn) positive NSPCs, as well as post-mitotic neurons and glial cells, experiments were performed to test whether expression of *cap-d3 dsRNA* driven by different, cell type-specific GAL4 drivers could recapitulate the loss of adult brain volume and/or adult head size seen with *eyGAL4, GMRGAL4*. *Cap-d3 dsRNA* expression driven by *repoGAL4*, or by *elavGAL4*, which respectively drive expression in glial cells or in post-mitotic neurons, had no effect on adult head size (Supp. Fig. 1a–d). However, when two different GAL4 drivers, *optixGAL4*, and *c253GAL4*, were used to drive expression of *cap-d3* dsRNA in regions of the central brain that contain Dpn+ NSPCs (Fig. 4b, c), adult male brain volumes were significantly decreased, as compared to controls

expressing *GFP dsRNA* (Fig. 4d, e). Interestingly, *optixGAL4* and *c253GAL4*-mediated Cap-d3 insufficiency did not affect adult female brain volumes (Supp. Fig. 2), again suggesting that microcephaly caused by Cap-d3 insufficiency in the developing brain, may be more severe in males. Additionally, while *eyGAL4*, *GMRGAL4*, and *optixGAL4* all drive expression in developing eye disks, *c253GAL4* does not[43], suggesting that *cap-d3 dsRNA* expression in eye disks is not required to cause microcephaly. It should be noted, however, that impaired development of eye disks can affect head development[44–47], and it is, therefore, possible that *eyGAL4*, *GMRGAL4*, and *optixGAL4*-driven expression of condensin dsRNAs in developing eye disks could contribute to the severity of the microcephaly observed. Finally, due to the fact that *eyGAL*, *GMRGAL4*, *optixGAL4*, and *c253GAL4* expression patterns all include regions containing Dpn+ central brain NSPCs, as well as Dpn+ optic lobe NSPCs, a *dpnGAL4* driver was used to drive expression of *cap-d3 dsRNA* primarily in optic lobe stem cells (Supp. Fig. 3a) and test whether Cap-d3 knockdown in these cells is sufficient

to decrease brain volume. Results showed no significant differences in adult male brain volumes between flies expressing *cap-d3 dsRNA* and controls (Supp. Fig. 3b). Therefore, these combined data suggest that Cap-d3 insufficiency in larval central brain NSPCs may be sufficient to cause microcephaly.

## Knockdown of Cap-d3 in the larval central brain increases NSPC death

Proliferation, differentiation, and programmed cell death are each required for the development of the brain in most organisms, including *Drosophila*, and must be tightly regulated to produce a normal, functioning adult organ (reviewed in refs. 1,48). To determine whether Cap-d3 insufficiency in third instar larval NSPCs affected any of these cellular processes, immunofluorescence analyses were first performed to detect markers of proliferation (phosphorylated Histone H3; PH3) and differentiation (Prospero; Pros). No differences in the number of Dpn+, PH3+ or Dpn+, Pros+ stem and progenitor cells were observed

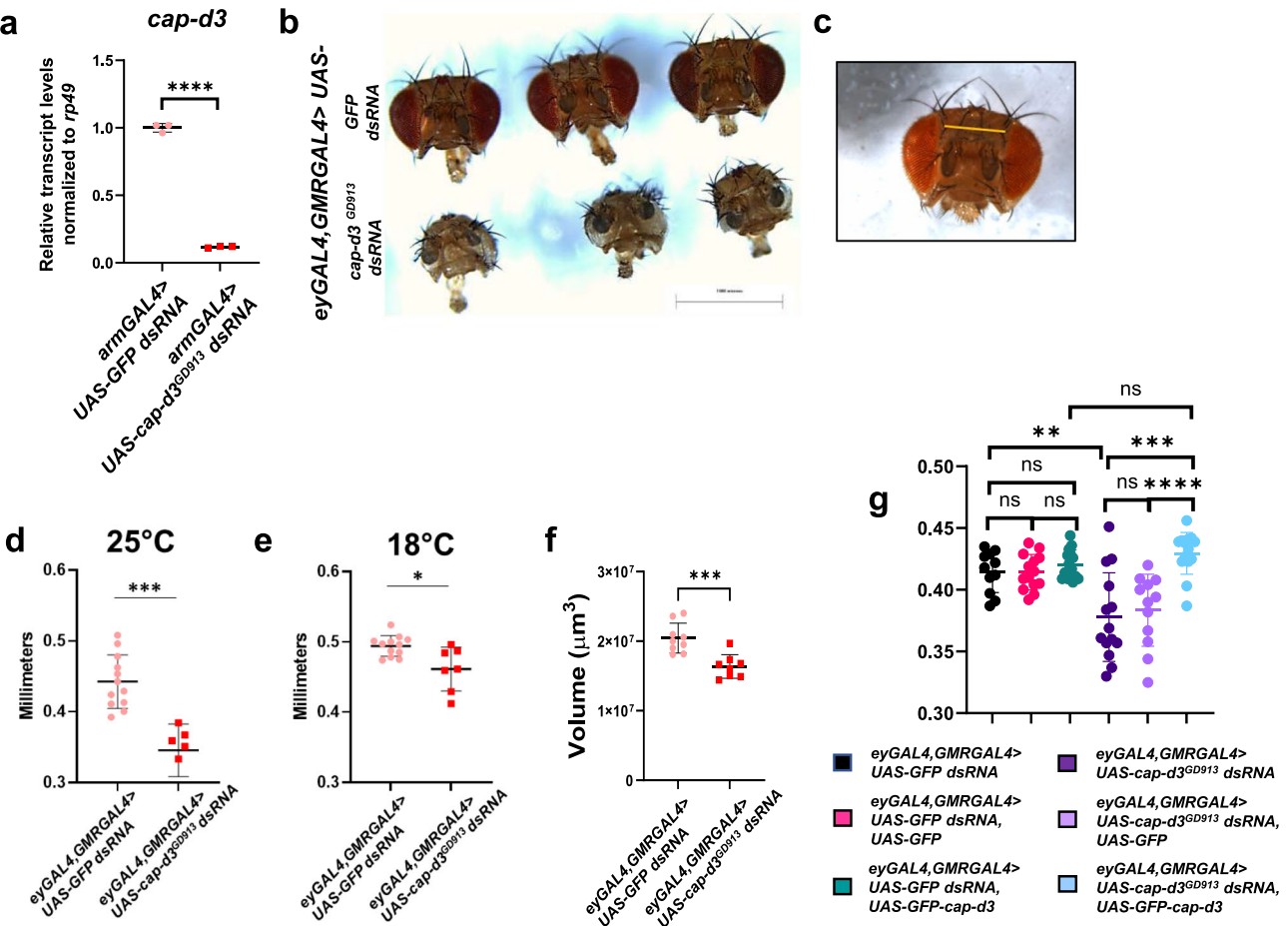

**Fig. 1 | Knockdown of condensin II subunit Cap-d3 results in smaller *Drosophila* adult heads and brains. a** qRT-PCR analyses of *cap-d3* transcripts were performed on cDNA generated from first instar larvae expressing control *UAS-GFP dsRNA* (pink) or *UAS-cap-d3 dsRNA* (red) under the control of *armGAL4*. Transcripts were normalized to the housekeeping gene, *rp49*. The chart shown includes technical replicates and is representative of two independent experiments. *P* values were determined by performing a two-tailed, unpaired *t*-test. **b** Adult female fly heads were dissected from flies expressing control *UAS-GFP dsRNA* (top row) or *UAS-cap-d3 dsRNA* (bottom row) under the control of *eyGAL4*, *GMRGAL4*. **c–e** The distance between two machrochaetes positioned between the eyes of adult flies was measured as a proxy for measuring adult head size. The region measured is shown in (**c**). Adult, female fly head sizes were measured on flies expressing control *UAS-GFP dsRNA* (pink) or *UAS-cap-d3 dsRNA* (red) driven by *eyGAL4, GMRGAL4* at 25 °C (**d**) or at 18 °C (**e**). Charts shown in (**d**) and (**e**) are

representative of two independent experiments; each data point represents a single fly. **f** Adult brain volumes were measured in female flies (1–3 days posteclosure) expressing control *UAS-GFP dsRNA* (pink) or *UAS-cap-d3 dsRNA* (red) driven by *eyGAL4, GMRGAL4* at 25 °C. The charts shown are representative of two independent experiments; each data point represents a single brain. **g** Adult head sizes were measured in female flies expressing different combinations of control *UAS-GFP dsRNA* or *UAS-cap-d3 dsRNA* and *UAS-GFP* or *UAS-GFP-cap-d3* under the control of *eyGAL4, GMRGAL4*. The results shown are the combined results of two independent experiments; each data point represents a single fly. For experiments in (**d–g**), *P* values were determined by performing two-tailed Mann–Whitney analyses. For **a**, *p* ≤ 0.0001. For **d**, *p* = 0.0001. For **e**, *p* = 0.0121. For **f**, *p* = 0.0010. For **g**, **p* = 0.0060, ***p* = 0.0001, ****p* ≤ 0.0001. NS not significant. For all experiments, error bars indicate standard deviations from the mean.

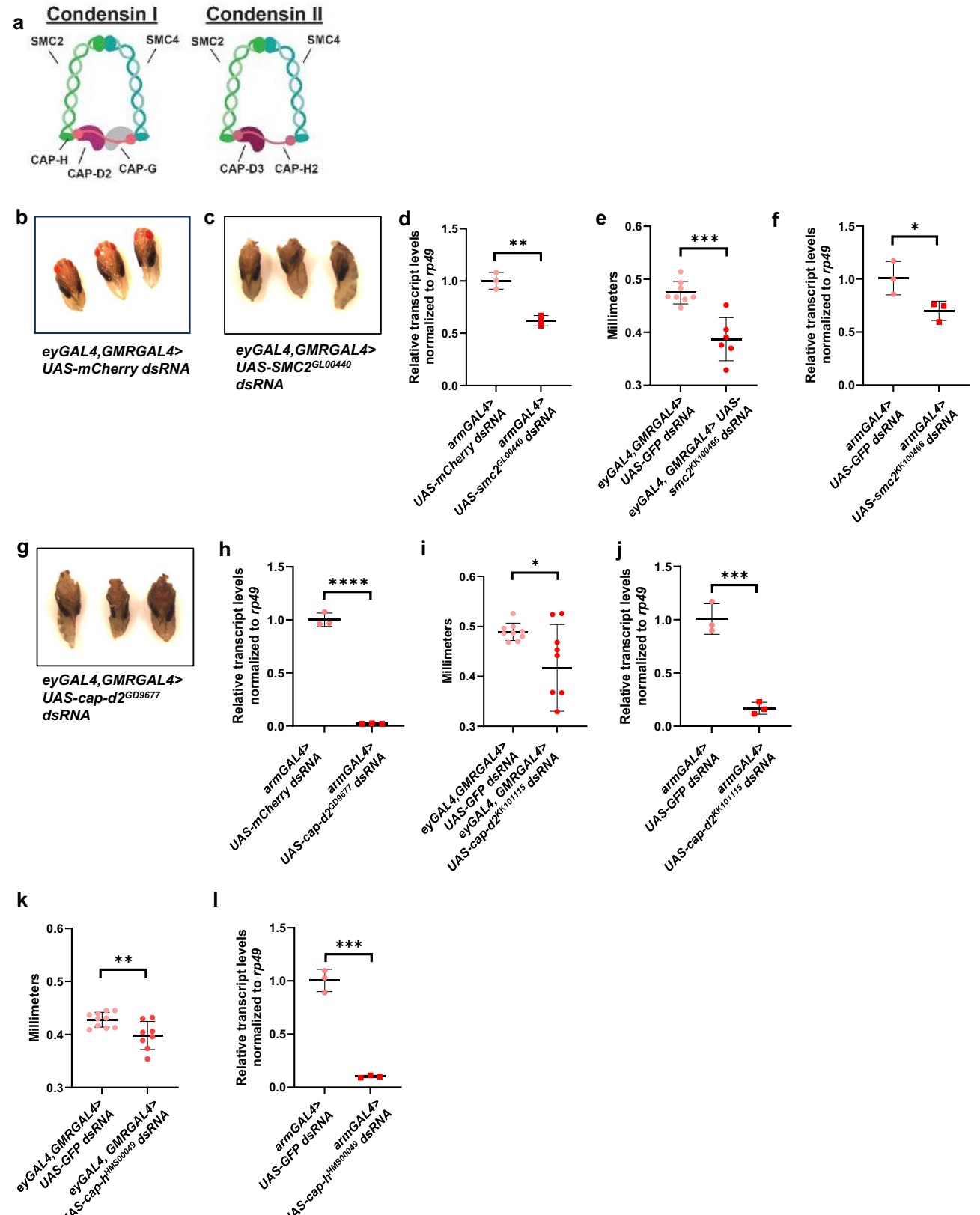

between *cap-d3 dsRNA* expressing brains and control brains (Fig. 5a, b). Total numbers of Dpn+ stem and progenitor cells present in the central brain also did not significantly change, although Cap-d3 insufficient brains trended toward a decrease (Fig. 5c). To assay for cell death, immunofluorescence analyses were performed to detect the effector caspase Death Caspase 1 (Dcp1) (Fig. 5d) in larvae expressing *cap-d3*

*dsRNA* or controls, *GFP dsRNA* or *eyGMR* (no RNAi expression). Results demonstrated significant increases in numbers of Dpn+, Dcp1+ stem and progenitor cells in brains expressing *cap-d3 dsRNA* (Fig. 5e). Furthermore, there was also an increase of Dcp1+, Dpn− cells positioned directly adjacent to the Dpn+ stem and progenitor cells were also observed in brains expressing *cap-d3 dsRNA* (Fig. 5f). Larval central

**Fig. 2 | Knockdown of multiple condensin subunits results in smaller adult heads. a** Diagram of the proteins that comprise the condensin I and condensin II complexes in *Drosophila melanogaster*. **b**, **c**, **g**) Female pupae expressing (**b**) *UAS-GFP dsRNA*, (**c**) *smc2 dsRNA*, or (**g**) *cap-d2 dsRNA* under the control of *eyGAL4, GMRGAL4* were dissected from pupal cases and imaged. **e**, **i**, **k** Adult, female fly head sizes were measured from flies expressing control (*UAS-GFP or UAS-mCherry*) *dsRNA* or dsRNA targeting the (**e**) *smc2*, (**i**) *cap-d2*, and (**k**) *cap-h* under the control of *eyGAL4, GMRGAL4*. The charts shown are representative of two independent experiments; each data point represents a single fly. *P* values were determined by performing two-tailed Mann–Whitney analyses. For (**e**), *p* = 0.0010.

For (**i**), *p* = 0.0376. For (**k**), *p* = 0.0067. **d**, **f**, **h**, **j**, **l** qRT-PCR analyses of **d**, **f** *smc2*, **h**, **j** *cap-d2*, and **l** *cap-h* transcript levels were performed on cDNA generated from first instar larvae expressing control (*UAS-GFP or UAS-mCherry*) *dsRNA* or dsRNAs targeting the respective condensin subunit mRNA under the control of *armGAL4*. Transcripts were normalized to the housekeeping gene, *rp49*. The charts shown include technical replicates of samples collected from four independent experiments (~200 larvae per experiment, combined into one sample per genotype). *P* values were determined by performing two-tailed, unpaired *t*-tests. For (**d**), *p* = 0.0021. For (**f**), *p* = 0.0421. For (**h**), *p* ≤ 0.0001. For all experiments, error bars indicate standard deviations from the mean.

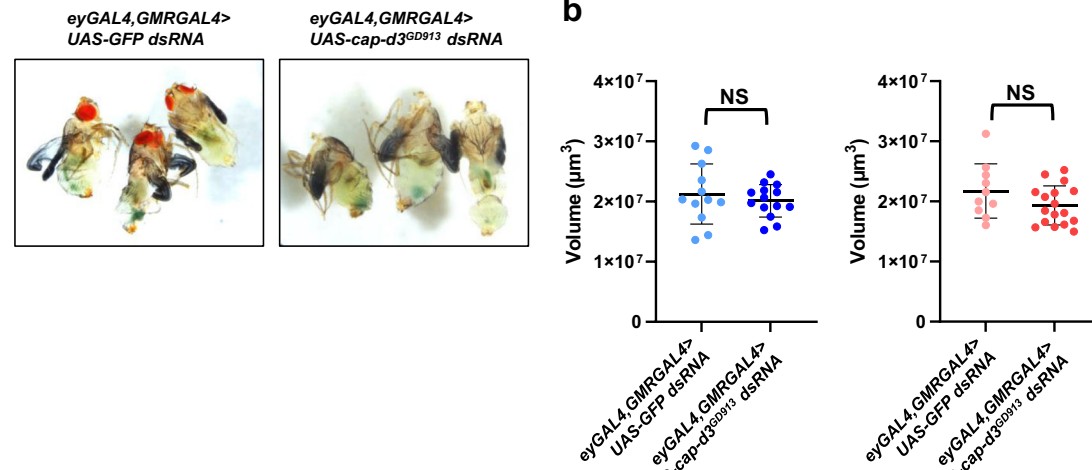

**Fig. 3 | Loss of volume in condensin-insufficient brains begins after the third instar larval stage. a** Larvae expressing control *UAS-GFP dsRNA* or *UAS-cap-d3 dsRNA*, under the control of *eyGAL4, GMRGAL4* were allowed to pupate for 72–96 h, at which time males were dissected and imaged. **b** The volumes of brains dissected from third instar larvae expressing control *UAS-GFP dsRNA* or *UAS-cap-d3 dsRNA*, under the control of *eyGAL4, GMRGAL4* were measured in males (left) and females (right). Results shown include larvae harvested from two independent experiments; each data point represents a single brain. *P* values were determined by performing two-tailed Mann–Whitney analyses. NS not significant. Error bars indicate standard deviations from the mean.

brain neural stem cells (Type I and Type II neuroblasts) and the ganglionic mother cells produced through asymmetric division of Type I neuroblasts are Dpn+. Not all of the cells produced through asymmetric division of neuroblasts are Dpn+, however, since Type II neuroblasts generate Dpn− intermediate progenitor cells (INPs) that only re-express Dpn following a maturation period[49,50]. Mature INPs then divide asymmetrically to self-renew and produce additional Dpn− cells. Therefore, the Dpn- cells neighboring the Dpn+ cells could either be direct descendants of neuroblasts, or they could be non-progenitor cells. Together, these data suggest that Cap-d3 insufficiency in the third instar larval central brain increases the number of dying NSPCs and the surrounding neighbors. Finally, to determine whether the increases in cell death were responsible for the reduced head sizes in flies expressing *cap-d3 dsRNA*, the *Drosophila* inhibitor of apoptosis 1 (Diap1) was overexpressed in combination with *cap-d3 dsRNA*. Diap1 overexpression caused adult female head sizes of *cap-d3 dsRNA* expressing flies to resemble the sizes of control flies expressing *GFP dsRNA* and Diap1 (Fig. 5g, blue compared to purple), thus rescuing the significantly smaller head sizes observed in flies expressing *cap-d3 dsRNA* alone (Fig. 5g, blue compared to red).

## Cap-d3 insufficiency increases retrotransposition in larval NSPCs

We previously published that condensin proteins repress RTE expression and mobility in *Drosophila* cells and tissues[26]. To determine whether Cap-d3 knockdown in third instar larval brains causes de-repression of RTE transcripts, qRT-PCR was performed to detect transcript levels of three separate RTE families, *mdg1*, *mdg4* (formerly known as *gypsy*), and *X-element* in third instar larval brains of larvae that were transheterozygous for a hypomorphic mutant *cap-d3* allele and a chromosomal deficiency spanning the region encompassing the *cap-d3* locus (*cap-d3^c07081/Df(2L)Exel7023*). Results demonstrated significant increases in transcript levels of all three RTE families in mutants, as compared to *w^1118* controls (Fig. 6a). Since RTE expression is the first step in the process of retrotransposition, we used the *gypsyCLEVR* reporter system[51] to test whether *mdg4* retrotransposition, in addition to expression, was also increased in *cap-d3 dsRNA* expressing larval brains. This reporter causes cells to express a membrane-localized GFP protein and a nuclear mCherry protein under the control of a UAS, immediately following retrotransposition of a *mdg4* RTE which is under the control of its own promoter[51]. To detect cells in which *mdg4* retrotransposition events had occurred, immunofluorescence analyses were performed on third instar larval brains with anti-mCherry antibody (pseudocolored green). Brains were also immunostained with antibody to Dpn (pseudocolored red). Unexpectedly, many mCherry-positive cells were observed in the central brains of control, *GFP dsRNA* expressing larval brains, and the majority of these cells either stained positive for Dpn or were adjacent to Dpn+ cells (Fig. 6b). Furthermore, *cap-d3 dsRNA* expressing brains exhibited significant increases in the number of cells in which *mdg4* retrotransposition events occurred (Fig. 6b, c), and the majority of these cells were also either Dpn+ or a neighbor of a Dpn+ cell. These increases were not a result of general RNAi pathway activation, since larval brains expressing the *gypsyCLEVR* reporter and GAL4 driver, without dsRNA, exhibited similar numbers of Dpn+, mCherry+ cells, as compared to larval brains expressing *GFP dsRNA*. As expected, no mCherry-positive

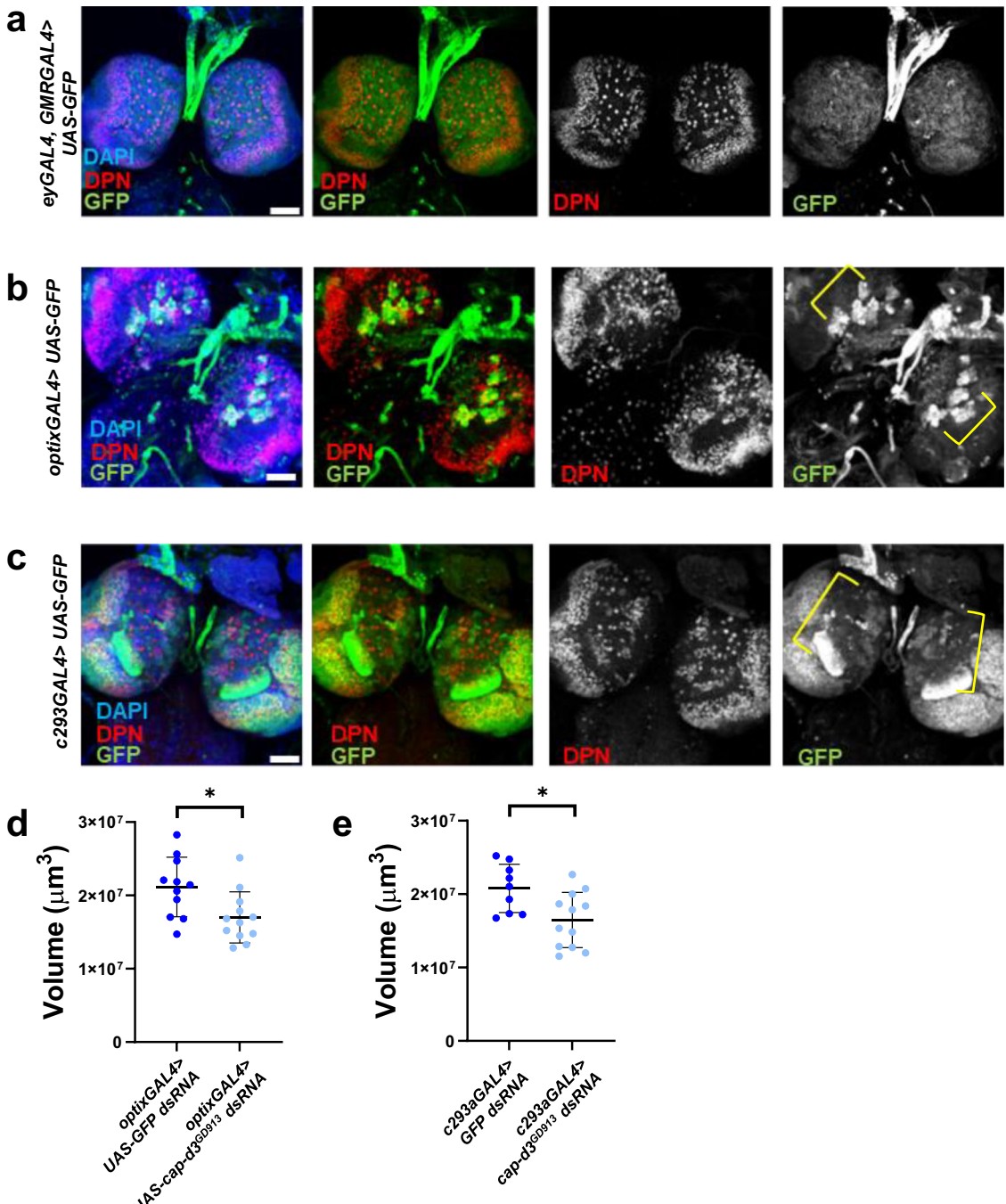

**Fig. 4 | Cap-d3 knockdown in the developing central brain reduces adult brain volumes. a–c** Third instar larval brains were dissected from male larvae expressing *UAS-GFP* under the control of **a** *eyGAL4, GMRGAL4,* **b** *optixGAL4,* or **c** *c253GAL4,* immunostained with antibodies to detect GFP (green) and stem cell marker, Dpn (red), and imaged using confocal microscopy. Maximum projections of z-stacks are shown. Nuclei are stained with DAPI (blue). All images were taken with 40x magnification. All images shown are representative of at least five biological replicates. Scale bar = 50 μm. **d, e** Adult brain volumes were measured in male flies expressing control *UAS-GFP dsRNA* or *UAS-cap-d3 dsRNA* driven by **d** *optixGAL4* or **e** *c253GAL4* at 25 °C; each data point represents a single brain. *P* values were determined by performing two-tailed Mann–Whitney analyses. For (**d**), *p* = 0.0156. For (**e**), *p* = 0.0278. Error bars indicate standard deviations from the mean. Results shown include adults harvested from two independent experiments.

cells were observed in larval brains expressing a *gypsyCLEVR* mutant[51] which was used as a negative control (Supp. Fig. 4). These data suggest that RTE expression and mobility is increased in Cap-d3 insufficient NSPCs, as well as in the neighboring cells. Interestingly, the increased percentages of cells exhibiting retrotransposition events in Cap-d3 insufficient larvae were returned to control levels when larvae were allowed to develop on food containing the Nucleoside Reverse Transcriptase Inhibitor (NRTI), Azidothymidine (AZT) (Fig. 6d). Results also

revealed that AZT does not completely abrogate retrotransposition events in developing control or cap-d3 insufficient brains. The most likely explanation for these results is that the increased retrotransposition events observed in Cap-d3 insufficient brains occur after the development of the first instar larval mouth hooks. Once the mouth hooks develop and larvae ingest AZT, this inhibits future retrotransposition events. Remaining retrotransposition events, detected by the gyspyCLEVR reporter in both Cap-d3 insufficient and control

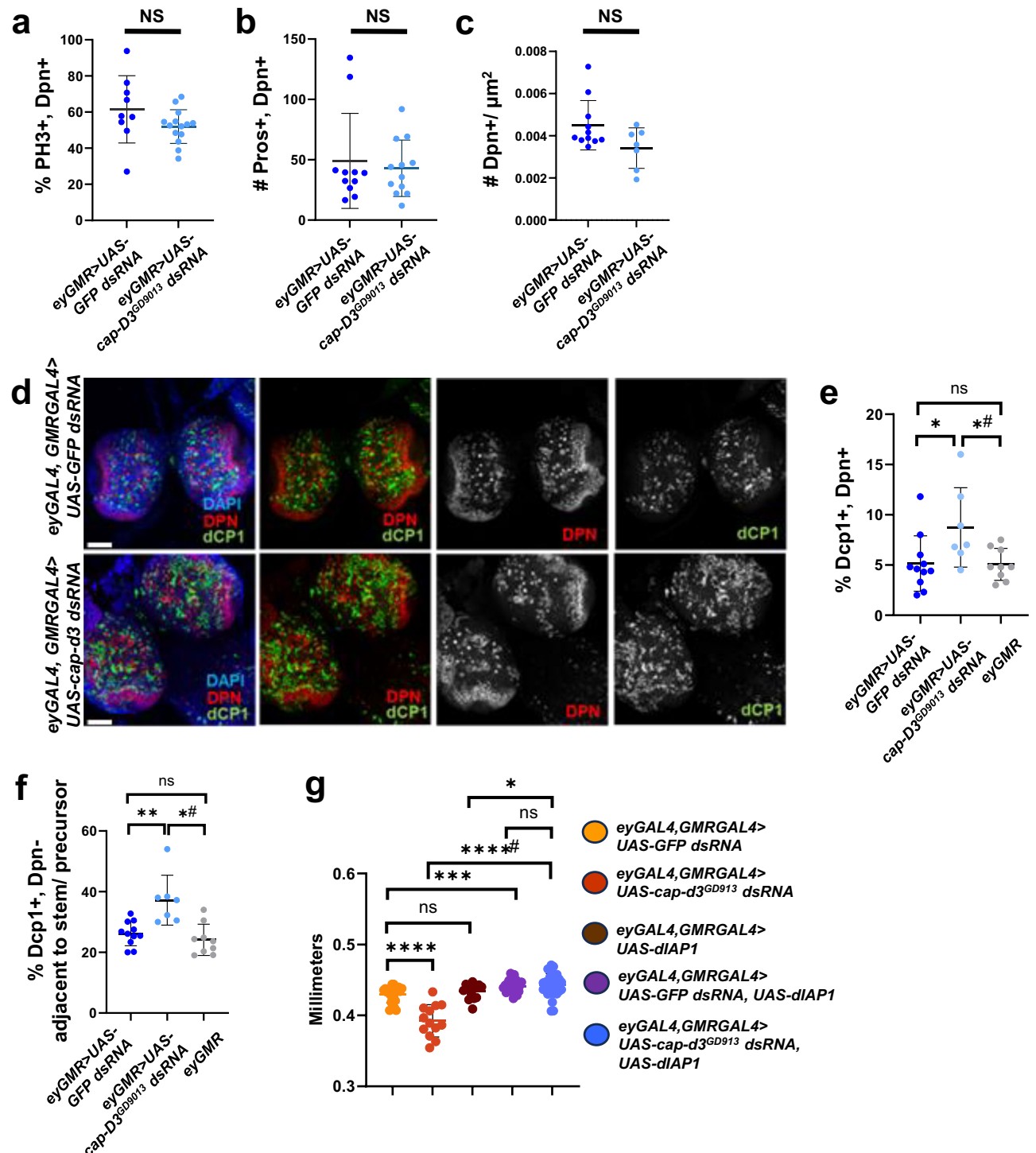

brains, would have occurred prior to this developmental stage, thus causing them to be unaffected by the addition of NRTIs to the food.

## NRTIs rescue microcephaly induced by Cap-D3 insufficiency

RTE expression and mobilization can lead to cell death in human cell lines[52], and in a cortical organoid-derived model of the early onset inflammatory disorder, Aicardi-Goutières syndrome[53]. De-repression/overexpression of retrotransposons in adult neurons of *Drosophila* models of neurodegenerative disease can also lead to cell death[54,55]. To determine whether inhibition of RTE activity could prevent the increased NSPC death in *cap-d3 dsRNA*-expressing larval brains, male larvae were raised in food containing 5 μM AZT or DMSO. *Cap-d3*

*dsRNA* expressing larval brains dissected from larvae developed on AZT, exhibited similar numbers of Dpn+, Dcp1+ NSPCs as compared to *GFP dsRNA* expressing larval brains dissected from larvae developed on AZT or DMSO (Fig. 7a). This was also true for Dpn− cells neighboring the Dpn+ cells (Fig. 7b). Additionally, cap-d3 transheterozygous mutants developed on DMSO also exhibited increased percentages of Dcp1+ NSPCs, and these percentages were decreased to levels resembling wild-type controls when larvae were developed on AZT (Fig. 7c).

The involvement of RTE expression and activity in causing reduced head sizes in flies expressing *cap-d3 dsRNA* under the control of *eyGAL4, GMRGAL4*, was also analyzed (Fig. 7d). Results

**Fig. 5 | Cap-d3 insufficiency increases cell death in the third instar larval central brain. a** Third instar larval brains were dissected from male larvae expressing control *UAS-GFP dsRNA* or *UAS-cap-d3 dsRNA*, under the control of *eyGAL4, GMRGAL4*, immunostained with antibodies to detect stem cell marker, Dpn, and mitotic cell marker, PH3, and imaged using confocal microscopy. The percentage of cells that stained positive for both Dpn and PH3 were quantified. **b** Immunostaining for Dpn and Pros, a marker of differentiation, was performed in male third instar larval brains. The number of cells that stained positive for both Dpn and Pros were quantified. **c** The number of Dpn-positive cells per square micrometer of larval central brain tissue in larvae expressing *cap-d3 dsRNA* or control, *gfp dsRNA* was quantified. For **a**–**c**, results shown include larvae harvested from two independent experiments; each data point represents a single brain. *P* values were determined by performing two-tailed Mann–Whitney analyses. NS not significant. **d** Immunostaining to detect Dpn and the cell death marker, Dcp1 was performed in male third instar larval brains expressing control *UAS-GFP dsRNA* (dark blue) or *UAS-cap-d3 dsRNA* (light blue), under the control of *eyGAL4, GMRGAL4*. Images were taken with a confocal microscope. Maximum projections of z-stacks are shown. Nuclei are stained with DAPI (blue). All images were taken with 40x magnification. Scale bar = 50 μm. **e** The percentage of cells that stained positive for both Dpn and Dcp1 in (**d**) were quantified. Additionally, larval brains expressing *eyGAL4* and *GMRGAL4* (gray; no RNAi control) were imaged and quantified. Each data point represents a single brain. **f** Dcp1+, Dpn− cells located adjacent to Dpn-positive cells in male third instar larval brains (experiments described in **d**, **e**) were quantified. Each data point represents a single brain. **g** Adult female head sizes were measured in female flies expressing different combinations of control *UAS-GFP dsRNA* or *UAS-cap-d3 dsRNA* and the *Drosophila* inhibitor of apoptosis, *UAS-diap1*, under the control of *eyGAL4, GMRGAL4*. For **e**–**g**, results shown include larvae harvested from two independent experiments; each data point represents a single brain (**e**, **f**) or a single fly (**g**). *P* values were determined by performing two-tailed Mann–Whitney analyses. For (**e**), *$p = 0.0212$ and *#$p = 0.0411$. For (**f**), *$p = 0.0013$ and *#$p = 0.0024$. For (**g**), *$p = 0.0422$, ***$p = 0.0004$, ****$p \leq 0.0001$, ****#$p \leq 0.0001$, NS not significant. For all experiments, error bars indicate standard deviations from the mean.

demonstrated that raising *cap-d3 dsRNA* expressing flies on food containing AZT prevented the significant decreases in adult head size observed in *cap-d3 dsRNA* expressing flies raised on DMSO, causing head size measurements to more closely resemble control *GFP dsRNA* expressing flies developed on DMSO or AZT (Fig. 7d). This was also true for flies developed on stavudine (d4T), another NRTI (Fig. 7e), and for *cap-d3* transheterozygous or heterozygous mutants developed on AZT, when compared to wild-type (*w^1118^*) controls (Fig. 7f). Together, these data suggest that unrestricted RTE expression may be responsible for the increased numbers of dying cells in Cap-d3 insufficient third instar larval central brains, and NRTI-mediated inhibition of RTEs can prevent both the increased cell death, as well as the resulting decreases in brain volumes and head size.

To determine whether the general upregulation of RTEs could result in similar phenotypes, *ago2* mutants were analyzed for levels of NSPC death in the larval central brain and for the development of microcephaly in the absence or presence of NRTIs. AGO2 is required in somatic tissues to repress RTE expression[56–59]. The *ago2^454^* allele was shown to carry a deletion of ~6 kb that removes the PAZ and PIWI domains of AGO2, resulting in a null allele[60]. The *ago2^Df(3L)BSC558^* allele is a deficiency allele harboring a much larger chromosomal deletion which has been mapped but is uncharacterized in relation to *ago2* expression. To confirm that flies expressing the *ago2^Df(3L)BSC558^* exhibit decreased *ago2* transcript levels, qRT-PCR was performed on RNA harvested from dissected intestines of *ago2^Df(3L)BSC558^* adults that were reared on food containing DMSO or AZT. Results demonstrated reduced *ago2* transcript levels, in comparison to wild-type, *w^1118^* controls (Supp. Fig. 5a), regardless of which food they were developed on. Interestingly, ago2 transheterozygous larval brains exhibited increased percentages of dying NSPCs, and this was rescued by allowing larvae to develop on food containing AZT (Supp. Fig. 5b). Furthermore, ago2 mutants also possessed smaller head sizes, as compared to wild-type flies, and this could also be rescued by ingestion of AZT (Supp. Fig. 5c). Combined, these results suggest that general upregulation of RTE expression and activity, independent of genetic background, may impair brain development and cause microcephaly in *Drosophila*.

## Discussion

Collectively, our results suggest: (1) Tissue-specific knockdown of condensin I and condensin II proteins in the central brain results in significant decreases in adult brain volume and adult head sizes, thus modeling microcephaly observed in patients with condensin loss of function mutations; (2) RTE expression and activity are upregulated in NSPCs of third instar larval central brains expressing decreased levels of condensin proteins, leading to increased numbers of dying cells; (3) The ingestion of NRTIs during development inhibits RTE activity, allowing larvae to develop into adults with normal head sizes. These findings suggest a working model where condensin I and condensin II restrict RTE expression and mobilization in NSPCs, thus preventing an overabundance of cell death in the third instar larval brain which would lead to decreased brain volumes, starting in the pupal stage, and ultimately cause reduced adult head sizes (Fig. 8).

The tissue-specific knockdown of Cap-d3 and other condensin subunits in the developing *Drosophila* brain, driven by *eyGAL4* and *GMRGAL4*, results in significant decreases in adult brain volumes and head sizes (Figs. 1, 2). Similarly, *cap-d3* mutants also exhibit decreased brain volumes and head sizes (Fig. 7), in addition to decreases in overall body weight[61]. This suggests that decreasing condensin subunit expression can have the universal effect of decreasing organ size in *Drosophila*, regardless of the cell type. *cap-h2* mutant mice also exhibit overall decreases in body weight, in addition to microcephaly[12]. Interestingly, germline condensin mutations identified in human microcephaly patients caused some, but not all, of the patients to exhibit significantly decreased stature, or even dwarfism[12]. It is perplexing that some germline mutations that cause decreased expression of condensin proteins, which function in every cell type, cause reductions in brain volume in some patients without affecting overall body size. LINE-1 RTEs can mobilize within mammalian neural stem cells[31,62,63], and LINE-1 and *Alu* insertions have been identified in adult human neurons and glia[64]. However, these insertions were not observed in fibroblasts from the same organism/individual[31,64]. Additionally, whole-genome sequencing of brain tissues revealed that the majority of somatic LINE-1 retrotransposition events resulted in the integration of the LINE-1 sequence into a preexisting L1 sequence[65]. Therefore, we hypothesize that RTE expression and activity may be higher in developing human brain tissues than in other somatic tissues within the body, and thus, the depletion of condensin protein increases RTE expression and activity to very high levels that cannot be tolerated in the brain. This may not happen in other organs where baseline RTE expression/activity levels are much lower, such that condensin insufficiency still results in higher RTE expression/activity, but it remains below a threshold that can still be tolerated by the cells. It is also possible that in some organisms, the mechanisms that regulate condensin expression/ activity may become dysregulated, specifically in the brain. The identity of the major mechanisms regulating condensin expression still remains largely unknown, and future studies aimed at elucidating these mechanisms in the brain will be important for understanding the tissue-specific vs. whole-body effects of condensin insufficiency.

Cap-d3 knockdown results in decreased pupal brain volumes, which in males, causes pupal heads to almost completely disappear and results in lethality (Figs. 2, 3). Females, however, do eclose, and exhibit reduced brain volumes and head sizes, but their heads are

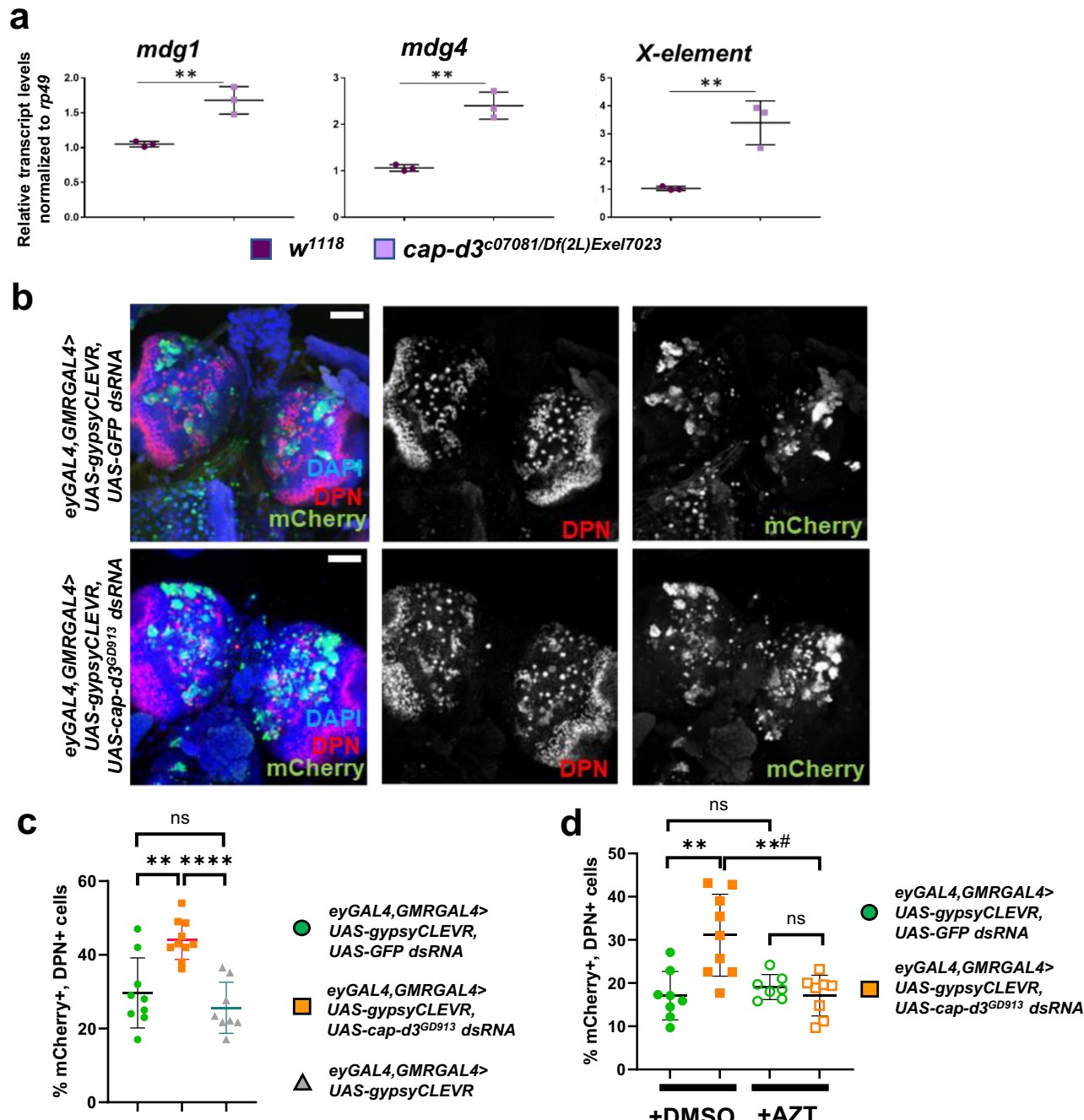

**Fig. 6 | RTE expression and activity are upregulated in Cap-d3 insufficient larval neuroblasts. a** qRT-PCR analyses of RTE (*mdg1, mdg4,* and *X-element*) transcripts were performed with cDNA generated from third instar larval brains dissected from *w[1118]* controls or *cap-d3* transheterozygous mutant larvae (150 brains per genotype; mixed sexes). Transcripts were normalized to the housekeeping gene, *rp49*. Results show the averages of three independent biological replicates. *P* values were determined by performing two-tailed, unpaired *t*-tests. For *mdg1*, ***p* = 0.0054, for *mdg4*, ***p* = 0.0015, and for *x-element*, ***p* = 0.0067. **b** Third instar larval brains were dissected from male larvae expressing *UAS-gypsyCLEVR*[51] and *UAS-GFP dsRNA* or *UAS-cap-d3 dsRNA* under the control of *eyGAL4, GMRGAL4*, immunostained with antibodies to mCherry, a reporter of gypsy retrotransposition (pseudocolored green), and stem cell marker, Dpn (red), and imaged using confocal microscopy. Maximum projections of z-stacks are shown. Nuclei are stained with DAPI (blue). All images were taken with 40x magnification. Scale bar = 50 μm. **c** The percentage of cells staining positive for both mCherry and Dpn were quantified from experiments

described in (**b**). Male larval brains expressing eyGAL4 and GMRGAL4-driven *UAS-gypsyCLEVR* were also analyzed as "no RNAi" controls. Results shown include larvae harvested from two independent experiments; each data point represents a single brain. **d** Third instar larval brains were dissected from male larvae expressing *UAS-gypsyCLEVR* and control *UAS-GFP dsRNA* (green circles) or *UAS-cap-d3 dsRNA* (orange squares), under the control of *eyGAL4, GMRGAL4*, immunostained with antibodies to detect mCherry and Dpn, and imaged using confocal microscopy. The percentage of cells that stained positive for both Dpn and Dcp1 were quantified. Flies were developed on food containing DMSO as a control (closed circles and squares), or on food containing 5 μM AZT (open circles and squares). Results shown include larvae harvested from two independent experiments; each data point represents a single brain. For experiments in (**c, d**), *P* values were determined by performing two-tailed Mann–Whitney analyses. For (**c**), ***p* = 0.0029, *****p* ≤ 0.0001, NS not significant. For (**d**), ***p* = 0.0037, ***#p* = 0.0039, NS not significant. Error bars indicate standard deviations from the mean.

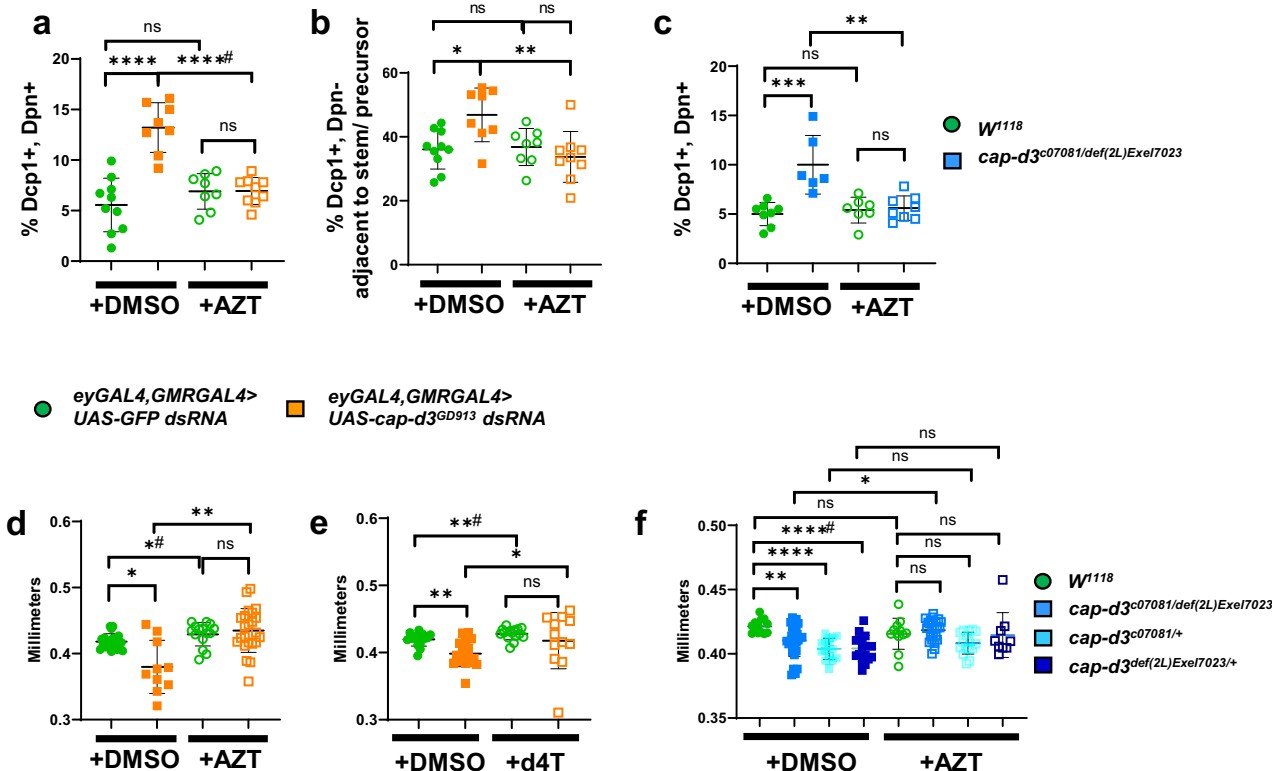

**Fig. 7 | Repression of RTEs prevents cell death in condensin insufficient larval neuroblasts and adjacent cells. a** Third instar larval brains were dissected from female larvae expressing control *UAS-GFP dsRNA* (green circles) or *UAS-cap-d3 dsRNA* (orange squares), under the control of *eyGAL4, GMRGAL4*, immunostained with antibodies to detect stem cell marker, Dpn, and cell death marker, Dcp1, and imaged using confocal microscopy. The percentage of cells that stained positive for both Dpn and Dcp1 were quantified. Larvae were developed on food containing DMSO as a control (closed circles and squares), or on food containing 5 μM AZT (open circles and squares). **b** Dcp1+, Dpn− cells located adjacent to Dpn-positive cells in female third instar larval brains (experiments described in (**a**)) were quantified. Results shown in (**a, b**) include larvae harvested from two independent experiments; each data point represents a single brain. **c** Experiments described in (**a**) were performed on brains dissected from *cap-d3* transheterozygous mutant (blue squares) male larvae or wild-type (green circles) male larvae. Results shown include larvae harvested from two independent experiments; each data point represents a single brain. **d, e** Adult female fly head sizes were measured on flies

expressing control *UAS-GFP dsRNA* (green circles) or *UAS-cap-d3 dsRNA* (orange squares) driven by *eyGAL4, GMRGAL4* at 25 °C. Flies were developed on food containing DMSO as a control (closed circles and squares), or on food containing (**d**) 5 μM AZT or (**e**) 5 μM d4T (open circles and squares). **f** Adult male head sizes were measured on wild-type flies (green circles), on *cap-d3* transheterozygotes (sky blue squares) or *cap-d3* heterozygotes (light blue and dark blue squares). Flies were developed on food containing DMSO as a control (closed circles and squares), or on food containing 5 μM AZT (open circles and squares). Results shown in (**d**–**f**) include adults harvested from two independent experiments; each data point represents a single fly. *P* values were determined by performing two-tailed Mann−Whitney analyses. For (*a*), ****$p \leq 0.0001$ and ****# $p \leq 0.0001$. For (**b**), *$p = 0.0434$, **$p = 0.0079$, NS not significant. For (**c**), **$p = 0.0013$, ***$p = 0.0007$, NS not significant. For (**d**), *$p = 0.0208$, *#$p = 0.0336$, **$p = 0.0014$. For (**e**), *$p = 0.038$, **$p = 0.0015$, **#$p = 0.0084$, NS not significant. For (**f**), *$p = 0.0181$, **$p = 0.0018$, ****$p \leq 0.0001$, ****#$p \leq 0.0001$, NS not significant. For all experiments, error bars indicate standard deviations from the mean.

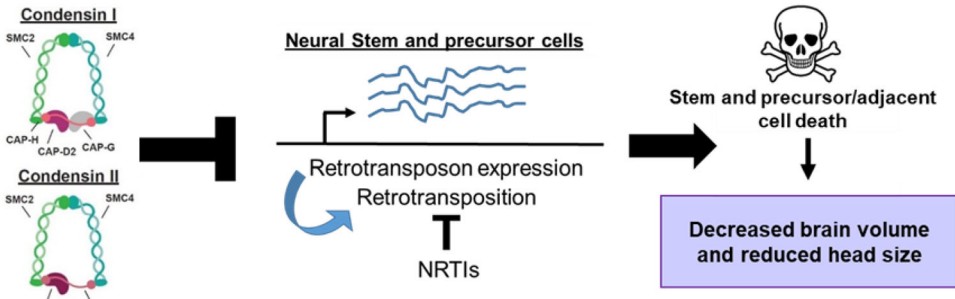

**Fig. 8 | Working model by which condensin complexes regulate brain development in *Drosophila*.** Condensin I and II restrict RTE expression and retrotransposition in neuroblasts and intermediate neural precursor cells in the

developing larval brain. This, in turn, restricts cell death of stem and precursor cells, as well as cells adjacent to stem/precursor cells, thereby allowing the brain to achieve proper volume by the late pupal stages.

more fully formed than their male counterparts (Fig. 1b vs 3a). Why do condensin-insufficient males exhibit more severe phenotypes? Different RTE families are expressed at different levels in males and females[66–69]. It is possible that the upregulation of a few specific RTE

families in condensin-insufficient *Drosophila* brains could be responsible for the observed phenotypes. Therefore, if those families naturally display higher expression levels in males, then further upregulation could have more severe effects. It will be interesting to

determine whether condensin insufficiency causes a global upregulation of all RTEs or specific RTE families in both males and females, and whether the upregulation is to an equal extent in both sexes.

Our results demonstrated that *mdg4* retrotransposition was permissible in control brains expressing *GFP dsRNA* or no RNAi (Fig. 6b, c). These *mdg4* retrotransposition events were primarily observed in Dpn + cells or Dpn− cells positioned within clusters surrounding the Dpn+ cells, which are, most likely, precursor cells (INPs) that directly descended from the Dpn+ neuroblast[49,50,70]. The fact that *eyGAL4* and *GMRGAL4* drives in a large area of the larval central brain (Fig. 4a), but retrotransposition events were only observed in NSPCs, suggests that these cell types may be the only cell types that permit RTE mobilization in the developing brain. It is also possible that eyGAL4 and GMRGAL4 drive expression at higher levels in NSPCs than in other retrotransposition-permissible cells in the central brain, potentially making it difficult to observe the reporter identifying retrotransposition events in these other cell types. Interestingly, NRTI ingestion did not prevent retrotransposition in developing control brains, although it did prevent the increased retrotransposition in *cap-d3* insufficient brains (Fig. 6d). We hypothesized that this was because the retrotransposition events that occurred in the control brains happened prior to the development of first instar larval mouth hooks. If this is true, then RTEs may play essential roles during early *Drosophila* brain development. RTEs can modulate mammalian gene expression through multiple mechanisms, including the introduction of transcription factor binding sites, regulation of 3D genome architecture, production of ncRNAs, and generation of a silenced chromatin environment (reviewed in ref. 71). In fact, lineage-specific enhancers that are enriched in insertions of a specific family of SINEs play important roles in mammalian neural development[72,73]. Recently, an elegant study determined that L1s are expressed in apical and basal progenitor cells and early-born neurons present in developing human fetal forebrain; silencing of a specific L1 insertion in cerebral organoids developed from these cells resulted in smaller size[74]. Therefore, it is possible that RTEs also regulate gene expression programs that control early brain development in *Drosophila*, and future experiments involving scRNA-seq of wild type, condensin mutant, and ago2 mutants may shed light on these questions.

Interestingly, *mdg4* retrotransposition events detected using the *gypsyCLEVR* reporter in the adult *Drosophila* brain, were observed at a much lower frequency than what we observed in the larval brain[51]. This suggests that cells which have undergone retrotransposition events during development must largely disappear during metamorphosis. Similar observations were recently reported in the *Drosophila* hindgut[75]. Our data show that unrestricted RTE expression and activity in condensin-insufficient or *ago2* deficient NSPCs results in increased cell death (Fig. 7 and Supp. Fig. 5). Combined with the possibility that cells harboring *mdg4* retrotransposition events seem to disappear during metamorphosis, we hypothesize that RTE expression and activity may contribute to the balance of proliferation and cell death that is necessary for the proper development of the brain. Programmed cell death is, in fact, necessary to eliminate post-mitotic neurons of specific neuroblast lineages which are generated during larval development and eliminated in early pupal stages[76], and it will be interesting to determine whether RTEs may play a role in these processes. Finally, RTE insertions in mammalian neural precursor cells occur frequently in genes associated with neuron-specific functions and can regulate the expression of those genes to drive neuronal somatic mosaicism[63,77]. Recent evidence demonstrated that the knockdown of condensin I subunit Cap-g in post-mitotic neurons affected the expression of both neuronal and non-neuronal genes[78]. It is, therefore, conceivable that retrotransposition of RTEs could promote somatic mosaicism in larval central brain NSPCs. However, if the majority of larval brain cells exhibiting retrotransposition events die during metamorphosis (as discussed above), then the effects of mosaic gene expression in larval neurons generated from the NSPCs would most likely be very transient.

While our findings suggest that increased RTE expression and activity is a potential mechanism driving condensin-insufficient NSPC death, the molecular events that activate apoptotic pathways downstream of RTE expression/activity are unresolved. Condensin protein insufficiency leads to increased levels of DNA damage in *Drosophila* cells, specifically within loci containing RTEs[26]. Decreased condensin expression in *Drosophila* larvae also results in increased transcription of antimicrobial peptides, which are major effectors of innate immune responses in *Drosophila*[79]. Both DNA damage, and increased immune signaling can be caused by the failure to repair DNA breaks at the site of RTE insertions and this can activate cell death pathways in mammalian cells[52,80–87]. Cytoplasmic DNAs generated by increased RTE expression and activity can also activate cGAS-STING and IFN signaling in mammalian cells[82,83], which could lead to cell death. While *Drosophila* possesses cGAS-STING homologs[88,89], their functions in innate immune signaling are still being studied, and we currently do not know whether they are required for the increased cell death observed in condensin-insufficient or ago2 deficient larval brains. Likewise, it will be interesting to determine in future studies whether RTE insertion or simply the expression of RTEs are required for the loss of brain volumes and smaller adult head sizes observed in these microcephaly models.

Finally, our studies invoke the question of whether the upregulation of RTEs is a common mechanism that could lead to microcephaly development in other genetic backgrounds. Interestingly, other microcephaly proteins (proteins encoded by genes identified to be mutated in microcephaly patients) have been associated with RTE expression or repression of virus[90–93]. Additionally, the microcephaly protein MCPH1 has been shown to physically and genetically interact with condensin proteins[94–97]. Thus, a more extensive investigation of potential roles for all known microcephaly proteins in repressing RTEs in the brain will be required to fully elucidate the generality of RTE-mediated mechanisms of microcephaly.

## Methods
### Fly maintenance
All *Drosophila melanogaster* stocks and crosses were maintained at 25 °C on standard dextrose medium, unless otherwise noted. To prepare food containing Nucleoside Reverse Transcriptase Inhibitors (NRTIs), 500 mL of standard dextrose medium was poured into four 1 L flasks. After allowing the media to cool below 60 °C, 50 µl of 50 mM AZT (Zidovudine)(Selleckchem), or d4T (Stavudine) (Selleckchem), dissolved in dimethyl sulfoxide (DMSO) were dispensed into flasks and mixed thoroughly, so that the final concentration of each drug in the food was 5 µM. About 50 µl of DMSO alone were also dispensed into flasks to prepare control media. Media was divided into 6 oz bottles and allowed to cool. NRTI-containing media was stored at 18 °C for no longer than 2 weeks before use.

### Fly stocks
**The following wild-type, mutant, and deficiency stocks were used in these studies.** *w[1118]* (Bloomington Stock Center 6326), *w[1118]*; *dCap-D3^{c07081/c07081}* (Harvard Exelixis collection generated in the *w[1118]* line[98], *w[1118]; Df(2 L)Exel7023/CyO* (Bloomington Stock Center 7797), *w[1118]; Df(3 L)BSC558/TM6C, Sb[1]* (Bloomington Stock Center 25120), *w[1118]; AGO2[454]/TM3, Sb[1] Ser[1]* (Bloomington Stock Center 36512)

**The following RNAi stocks were used in these studies.** *UAS-GFP dsRNA* (P{UAS-GFP.dsRNA.R}142; Bloomington Stock Center 9330), *UAS-cap-d3 dsRNA* (w1118; P{GD913}; VDRC stocks 29657 and 9402), *UAS-smc2-dsRNA* (P{KK100466}VIE-260B; VDRC stock 103406 and P{TRiP.GL00440}attP40; Bloomington Stock Center 35602), *UAS-cap-d2 dsRNA* (P{KK101115}VIE-260B; VDRC stock 108289 and *w1118*;

*P{GD9677}v33424*; VDRC stock 33424), *UAS-cap-h dsRNA (y1 sc* v1 sev21; P{TRiP.HMS00049}attP2*; Bloomington Stock Center 34068).

**The following GAL4 and UAS stocks were used in these studies.** *armadillo GAL4 (w[\*]; P{w[+mW.hs]=GAL4-arm.S}4a P{w[+mW.hs] =GAL4-arm.S}4b;* Bloomington Stock Center 1561 that has lost the balancer), *eyGAL4, GMRGAL4* (generous gift from Dr. Bob Johnston at Johns Hopkins University), *UAS-GFP.nls (P{UAS-GFP.nls}14;* Bloomington Stock 4775), *UAS-GFP-CAP-D3*[61], *tubPGAL4* (Bloomington Stock 5138), *repoGAL4* (generous gift from Dr. Heather Broihier at Case Western Reserve University), *elavGAL4* (Bloomington Stock Center 8765), *c253GAL4 (w[1118]; P{w[+mW.hs]=GawB}C253;* Bloomington Stock Center 6980), *dpnGAL4* (Bloomington Stock Center 47456), *optixGAL4 (w[1118]; P{y[+t7.7] w[+mC]=GMR30D11-GAL4}attP2;* Bloomington Stock Center 48098), *UAS-gypsyCLEVR* and *UAS-gypsy-CLEVR*[ΔPBS][51].

### RNA analysis by quantitative reverse transcriptase PCR (qRT-PCR)

Tissues were dissected, submerged in Trizol (Life Technologies) in a microcentrifuge tube, and homogenized using a Pellet pestle cordless motor (Kimble) and disposable Kontes pestle. For the first instar larvae collection, crosses (30 virgins and 15 males) were set up in bottles, and females were allowed to lay eggs onto grape agar plates, overnight. Grape agar plates were stored in the 25 °C incubator for 24 h, after which time, a paintbrush was used to collect larvae into a 100 μm nylon mesh cell strainer (Fisher). Larvae in cell strainers were submerged in 1X PBS, twice, and then submerged in 7% bleach in distilled water for 3 min, at room temperature. Larvae in cell strainers were then submerged in fresh 1X PBS for 30 s, and this was repeated four times. Larvae were transferred to microcentrifuge tubes, the remaining PBS was removed, and 50 μl Trizol was added to the tube. RNA was extracted during the standard Trizol protocol, according to the manufacturer.

Extracted RNAs were then treated with RNase-free DNase in buffer RDD (Qiagen) prior to further purification using an RNeasy Mini Kit (Qiagen). An aliquot of cDNA was generated from 0.5 to 2 μg of total RNA using the TaqMan Reverse Transcription Reagents (Applied Biosystems) and an oligo-dT(16) primer (Invitrogen). Quantitative RT-PCR was performed using the Roche Lightcycler 480 to amplify 15 μL reactions containing .5 μL of cDNA, 0.5 μL of a 10 μM primer mix, and 7.5 μL of Fast Start SYBR Green Master Mix (Roche). Each reaction was performed in triplicate. Crossing point (Cp/ Ct) values were determined using the Roche LightCycler 480 Absolute Quantification Second Derivative Analysis software. Relative quantitation of transcript levels was then performed using the delta-delta Ct method ($2^{-\Delta\Delta Ct}$), where the Ct values of a reference gene (*rp49*) in each sample are subtracted from the Ct values of the gene of interest to create a ΔCt value for each sample. The ΔCt is compared to a control sample to generate a ΔΔCt value for each sample. Following the calculation of $2^{-\Delta\Delta Ct}$ for each sample, triplicates were averaged. The sequences of oligos used in the qRT-PCR studies are listed below:

dCap-D3 FW: 5′-GGCGAATCATCAGCACCCTGC-3′
dCap-D3 RV: 5′-CAGGCATCGGTAGCCATGGAC-3′
dSMC2 FW: 5′-CGCGTAAGGTGCGTGGTTTG-3′
dSMC2 RV: 5′-GGGAATCAAGTGACACGACGCTG-3′
dCap-D2 FW: 5′-GGGAATCAAGTGACACGACGCTG-3′
dCap-D2 RV: 5′-CCATGCGAAGCGGCAAAAGTTCG-3′
dCap-H FW: 5′-GGCCTCACCCAGATGAACGCC-3′
dCap-H RV: 5′-TCGTCTCCTCCGGCACAGCG-3′
rp49 FW: 5′-TACAGGCCCAAGATCGTGAAG-3′
rp49 RV: 5′-GACGCACTCTGTTGTCGATACC-3′
ago2 FW: 5′-CCCCAACTCCATTGTCTGAACGTTG-3′
ago2 RV: 5′-CTTGCGCTTTCGCACGTTCGTC-3′
mdg4 FW: 5′-GTTCATACCCTTGGTAGTAGC-3′

mdg4 RV: 5′-CAACTTACGCATATGTGAGT-3′
mdg1 FW: 5′-AACAGAAACGCCAGCAACAGC-3′
mdg1 RV: 5′-TTTCTGATCTTGGCAGTGGA-3′
X-element FW: 5′-GCCAGCCTGCAACAGGTTGAAG-3′
X-element RV: 5′-CTCTGGCGCACAATGACTTCGG-3′

### *Drosophila* adult head measurements

Adult *Drosophila* were anesthetized with $CO_2$ and a razor blade was used to separate heads from bodies. Heads were stored at 4 °C until ready for imaging, at which point they were transferred to slides covered with double-sided sticky tape. A Leica MZ 16FA stereoscope with Leica imaging software was used to obtain images. Adobe Photoshop was used to measure the distance between two machrochaetes positioned between the eyes, depicted in Fig. 1c. This pair of machrochaetes was used in measurements for all figures. Pixels were converted to μm, based on the magnification used.

### *Drosophila* adult and larval brain measurements

To dissect adult brains from flies that were between 1 and 3 days post-eclosure, *Drosophila* heads were separated as described above and placed in cold PBS in a well of a nine-well, glass, dissection plate (Corning). Forceps were used to peel the cuticle and eye tissue away from the brain. Brains were fixed in 4% Paraformaldehyde/ TritonX-100 solution (4% paraformaldehyde made from 16% paraformaldehyde aqueous solution, EM grade (Electron Microscopy Sciences), 0.25 M EGTA, pH 8.0, 1X PBS, 0.5% TritonX-100) for 25 min at room temperature. Brains were then incubated in Vectashield with DAPI (Vector Laboratories) overnight, and placed onto slides containing tiny shards of crushed coverslips and vacuum grease to prevent squashing of brain tissue when coverslips were placed over the top of the slides. Brains were imaged with a Leica SP5 confocal/multi-photon microscope. Z stacks were processed using Leica Imaging software. Volocity imaging software (Quorum Technologies) was used to measure the volumes of individual brains; 3D images were analyzed using the Volocity "measurements" and "find objects" features, with the threshold adapted to use "intensity", and a minimum object size of 1,000,000 μm³. The population of Interest was selected for measurement.

To dissect larval brains, third instar larvae were placed into wells of a nine-well, glass, dissection plate (Corning). Larval brains containing the central brain, optic lobes, and ventral nerve cords were removed from the larvae using forceps. Fixation, slide preparation, and imaging were performed as described above for adult brains.

### Immunostaining of *Drosophila* larval brains

All tissues were fixed in 4% Paraformaldehyde/ TritonX-100 solution (described above) for 25 min, with rocking, at room temperature. Tissues were washed three times with 0.1% PBS-Triton and then incubated in blocking buffer (0.1% PBS-Triton/1%BSA) for 1 h at room temperature, with rocking. Tissues were then incubated with primary antibodies in a blocking buffer, overnight at 4 °C, with rocking. The following day, tissues were washed three times with 1X PBS at room temperature, with rocking. Tissues were then incubated with secondary antibodies for 1–2 h at room temperature, with rocking. Tissues were mounted in Vectashield with DAPI (Vector Laboratories), prior to imaging. Brains were imaged with a Leica SP5 confocal/multi-photon microscope. Z stacks were processed using Leica Imaging software. Cells positive for DPN, PH3, DCP1, and/or Pros were counted by hand. The total surface area of larval central brain tissues was measured using Image J.

### Antibodies

Primary antibodies used included anti-deadpan (Abcam ab195174; 1:100), anti-GFP (Invitrogen A10262; 1:200), anti-DCP1 (Cell Signaling 9578; 1:200), anti-phospho-histone H3 ser10 (Cell Signaling 9701; 1:500), anti-prospero (Developmental Studies Hybridoma Bank MR1A;

1:1000, and anti-mCherry (Cell signaling 43590; 1:200). All secondary antibodies were used at a dilution of 1:500 and included Goat anti-Chicken IgY (H + L) Secondary Antibody, Alexa Fluor™ 488 (Thermo Fisher A-11039), Goat anti-Rabbit IgG (H + L) Cross-Adsorbed Secondary Antibody, Alexa Fluor™ 568 (Thermo Fisher A-11011), Donkey anti-Rabbit IgG (H + L) Highly Cross-Adsorbed Secondary Antibody, Alexa Fluor™ 647 (Thermo Fisher A-31573), Donkey anti-Rabbit IgG (H + L) Highly Cross-Adsorbed Secondary Antibody, Alexa Fluor™ 488 (Thermo Fisher A-21206), Goat anti-Mouse IgG (H + L) Cross-Adsorbed Secondary Antibody, Alexa Fluor™ 488 (Thermo Fisher A-11001), Goat anti-Mouse IgG (H + L) Cross-Adsorbed Secondary Antibody, Alexa Fluor™ 568 (Thermo Fisher A-11004), and Goat anti-Mouse IgG (H + L) Cross-Adsorbed Secondary Antibody, Alexa Fluor™ 647 (Thermo Fisher A-21235).

## Statistics and data analysis

All Data analysis and statistical calculations were performed using GraphPad Prism software, version 10.

## Reporting summary

Further information on research design is available in the Nature Portfolio Reporting Summary linked to this article.

## Data availability

Further information and requests for resources and reagents should be directed to and will be fulfilled by the corresponding author, Michelle Longworth (longwom@ccf.org). Source data are provided with this paper.

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

## Acknowledgements

We thank members of the Longworth laboratory for their insightful discussions and advice during the performance of experiments and the preparation of the manuscript. We thank Dr. Judy Drazba in the Lerner Research Institute Imaging Core for assistance with imaging. We thank Dr. Josh Dubnau (Stonybrook University), Dr. Bob Johnston (Johns Hopkins University), Dr. Heather Broihier (Case Western Reserve University), and Dr. Pam Vanderzalm (John Carroll University) for their generous donations of fly stocks. Stocks obtained from the Bloomington *Drosophila* Stock Center (NIH P40OD018537) were used in this study. Transgenic fly stocks were also obtained from the Vienna *Drosophila* Resource Center (VDRC, www.vdrc.at). This work was supported by a Lisa Dean Moseley Foundation Award to M.S.L. and by a postdoctoral fellowship award F32HD114499 from NIH/NICHD.

## Author contributions

B.I.C., M.J.T. and M.S.L. conceived and designed the study and experiments. B.I.C., M.J.T., J.Ri., J.Ru., S.T., T.W. and M.S.L. performed experiments. B.I.C., M.J.T., J.Ri., J.Ru. and M.S.L. analyzed data. B.I.C., M.J.T. and M.S.L. interpreted the data. M.J.T. and M.S.L. wrote and edited the manuscript.

## Competing interests

The authors declare no competing interests.
