## [Peer Review File · Nature Communications]

Condensin-mediated restriction of retrotransposable element facilitates brain development in *Drosophila melanogaster*.Editorial Note: Parts of this Peer Review File have been redacted as indicated to remove third-party material where no permission to publish could be obtained.

REVIEWER COMMENTS

Reviewer #1 (Remarks to the Author):

This manuscript reports significant new insight into the contribution of RTEs to normal brain development and to mechanisms that may underlie microcephaly. These authors have previously reported that condensin plays a role in RTE regulation, and here they develop this idea into a novel hypothesis regarding the role of RTEs in mediating condensin-related microcephaly using a fly model. These findings also fit within a rapidly expanding literature on the contribution of RTEs and ERVs to both normal biology and to disease.

I am generally quite enthusiastic about this study.

I think that for the most part the experiments are well designed, and yield robust results that support the conclusions. I also think that the findings themselves will be of interest to a broad audience. I do have a few scientific critiques that I think the authors should address. These likely require some modest new experimentation, although I am open to hearing responses from the authors if they disagree (but I'll need some convincing).

Major:

My only major critique of this study is that the genetics is not fully compelling (yet). For almost all of the experiments, the authors use either a dsRNA against cap-d3 or a hypomorphic allele. As geneticists, we typically demand to see the convergence of two independent approaches in order to conclude that disruption to a particular gene explains the phenotypic effects. This can be failure to complement between two independently derived alleles, it can be two different RNAi lines that target different sequences in the same transcript, it can include a combination of mutation and RNAi, or mutations with transgenic rescue of key phenotypes. The problem I see here though is that the mutant allele used is dominant or at least semi dominant. This precludes use of true complementation testing. There is one experiment in which the authors use this allele over a Df, so a complementation test, but that is only in the context of RNA expression, not biological phenotypes. This makes me a bit uncomfortable. I would like to see either an independent RNAi line, or a full complementation test for some of the microcephaly or cell death phenotypes. If the homozygous state is sufficiently more severe than the heterozygous, then this could include 4 groups: wild type, Df/+, hypomorph/+, Df/hypomorph. If this supports the conclusion that this gene drives the effects on RTE (E.G. CLEVR) and cell death, and microcephaly, I would be more comfortable. Alternatively, a second RNAi, or a demonstration that hypomorph/Df precludes further impact of the RNAi could suffice. I think I would be comfortable if such 'genetic convergence' were better established for at least a few of the key phenotypes.

Related: In Figs 2B and C, the effects on head size are shown for smc2/+ or cap-d2/+. But in both of these panels, the "+" is a balancer (Cy0 or TM3). Thus formally, these effects on head size could be contributed by the balancer chromosome/cap-d2. This experiment should be re-done without the presence of balancer (just cross the mutant/balancer to a wild type and take the non balancer heterozygous progeny and compare them to wild type controls).

minor:

page 3, top: 'statistically no different' is an odd phrasing. maybe not statistically significant or no difference or something like that.

Authors should acknowledge the caveat that some of the differences in phenotype between different Gal4 lines could be due to differences in expression level of the RNAi.

page 20, typo noticed: "mobility in increased in" probably should read "mobility is increased in".

Finally, i very much appreciated the discussion section, which generally speaking is interesting and scholarly. The idea that CLEVR labeled larval neurons are lost normally during development, and that this could contribute to normal cell death is very interesting, and appropriate to speculate on. I do think though that the authors should discuss whether or not the effects of RTEs requires integration of de novo copies, just DNA damage from attempts at replication, or simply inflammatory effects from expression of dsRNAs or cDNAs. There is some literature that speaks to this, e.g. on LINE-1 elements driving cGAS/STING, on ERVs driving inflammation, and on ERVs causing DNA damage (e.g. in Drosophila neurodegeneration context).

Reviewer #2 (Remarks to the Author):

Crawford and colleagues have shown that dCap-D3 (a component of the condensin complex) is important for silencing retrotransposable elements (RTEs) in the developing brain of Drosophila. They show that perturbation of dCap-D3 results in the expression and activity of RTEs that can lead to reduced neurogenesis and smaller brains. Their experiments demonstrate that the increased RTE activity results in cell death and that by inhibiting cell death, the reduction in brain size was prevented. The most striking result is that the phenotypes can be rescued by feeding the larvae a drug that inhibits reverse transcriptases, and thereby RTE activity.

This is an important study highlighting the importance of condensin in repressing RTEs during development, and that this may be a key mechanism by which certain microcephaly phenotypes manifest. It also provides a very good model system for future research.

Overall, the investigations are thorough and well controlled. Also, the methods are sufficiently detailed.

There are some points below that should be addressed before publication:

- Should describe in more detail the expression patterns of ey-GAL4 and GMR-GAL4 in other tissues. For example, ey-GAL4 is expressed in the developing eye disc and phenotypes here could impact on the measurement taken of the adult heads.
- Scale bars in images should be included, especially when the zoom is obviously different within the same figure (e.g. Figure 4B).

- Is cell death occurring in ganglion mother cells (GMCs) but not immature neurons? This would be good to clarify. Could be tested by staining for DCP1, Pros and Elav.

- Figure 6 – it is surprising that RTE elements are active in the control GFP dsRNA experiments. Is general activation of RNAi machinery resulting in RTE expression? Testing whether this happens with a different RNAi control and in other developing tissues would be helpful for this study and more generally for RNAi studies. Related to this (Figure 7) - What about Dpn+ Dcp1+ numbers in brains not expressing any RNAi? Is generic RNAi also increasing apoptosis?

- Discussion – ‘driven by eyGAL4, GMRGAL4, results in significant decreases in adult brain volumes and head sizes, without affecting wing size (Fig. 1-3).’ Isn’t this just because GAL4 isn’t expressed there? What point is being made here?

- ‘The fact that eyGAL4, GMRGAL4 drives in a large area of the larval central brain (Fig. 4A), but retrotransposition events were only observed in NSPCs, suggests that these cell types may be the only cell types that permit RTE mobilization in the developing brain.’ – could this not just be due to differences in the level of GAL4 expression?

- ‘Recent evidence demonstrated that knockdown of condensin I subunit Cap-g in postmitotic neurons affected the expression of a subset of neuronal genes6.’ – It also results in the ectopic expression of non-neuronal genes – could this be linked to RTE expression?

Reviewer #3 (Remarks to the Author):

In this manuscript, Crawford et al. investigate the relationship between loss of condensin function, transposable element dysregulation, and abnormal brain development in *Drosophila*. They show that TEs are upregulated upon condensin knockdown, and that cell death via apoptosis also occurs. Condensin subunit knockdown is also shown to result in decreased brain and head sizes. These results are interpreted as a causal link between TE activity and microcephaly. Although the research question is compelling and some interesting results are presented, there are major issues with the internal logic and data interpretation central to the claims of this paper.

1. The first main conclusion of the paper, as outlined in the first paragraph of the discussion, is that tissue-specific knockdown of condensin I and II in the brain causes microcephaly. However, it is noted elsewhere in the paper that ubiquitous loss of condensin function causes a general reduction in organism size, and that this reduction is more prominent in male flies. Unless there is some evidence for brain-specific loss of condensin activity occurring naturally (in *Drosophila* or any organism), the purported link between condensin loss and microcephaly is entirely artificial.

2. The second major conclusion of the paper is that TE expression and activity downstream of condensin loss “leads to” increased cell death. This claim implies a causal relationship between TE expression/activity and apoptosis, but no evidence is supplied to support causation. Indeed, Ago2 mutant flies are used as a model of TE dysregulation in this manuscript. Do these flies exhibit increased apoptosis in the brain and decreased head sizes? If not, it is much more likely that cell death and abnormal brain development lie downstream of another function of condensin, aside from its role in TE suppression. However, there is no exploration of any other role for condensins in preventing apoptosis.

3. The third major conclusion is that NRTI ingestion during development inhibits TE retrotransposition as well as TE expression, and that this re-establishment of TE control results in normal head sizes in adult flies. This claim is highly problematic. NRTIs are reverse transcriptase inhibitors, and have not, to my knowledge, been demonstrated previously to inhibit TE expression. While NRTIs at very high doses can inhibit DNA polymerases, they are not characterized as RNA polymerase inhibitors. There is no explanation given by the authors as to why NRTIs might specifically repress TE expression, but not the expression of cellular genes (eg Ago2, which is shown in Sup Fig 6 to be unperturbed by AZT treatment). If NRTI treatment does, by some unknown mechanism, directly and specifically inhibit TE expression in addition to reverse transcription, it would be extremely interesting to the TE field. NRTIs are often used to support claims that retrotransposition or the generation of TE cDNAs has some effect on cellular function or disease state, and the demonstration that they actually impact TE expression would change the interpretation of numerous previous results. Unfortunately the claimed reduction in TE expression by NRTIs is only demonstrated explored in the intestines of ago2 mutant flies by qPCR, and is not demonstrated in condensin deficient brains exhibiting apparent rescue of head size by NRTIs. It therefore is equally likely that head size rescue is a side-effect of NRTI administration unrelated to TE expression or retrotransposition.

Response to Reviewer Comments

General Summary of changes:

We would like to thank the reviewers for their time and suggestions, which we believe have helped us to greatly improve this manuscript. We have addressed all reviewers' questions and comments, where possible, adding several new experiments, and changing the text in several places. Our detailed responses are listed below.

Response to comments raised by Reviewer #1:

Comment 1) I would like to see either an independent RNAi line, or a full complementation test for some of the microcephaly or cell death phenotypes. If the homozygous state is sufficiently more severe than the heterozygous, then this could include 4 groups: wild type, Df/+, hypomorph/+, Df/hypomorph. If this supports the conclusion that this gene drives the effects on RTE (E.G. CLEVR) and cell death, and microcephaly, it would be more comfortable. Alternatively, a second RNAi, or a demonstration that hypomorph/Df precludes further impact of the RNAi could suffice. I think it would be comfortable if such 'genetic convergence' were better established for at least a few of the key phenotypes.

RESPONSE: We have now added experiments to quantitate the levels of cell death in the neural stem and precursor cells (NSPCs) of the third instar larval central brain in *cap-d3* transheterozygous mutants, as compared to wild type (Fig. 7C). These experiments were also performed in food containing AZT or control, DMSO. Similar to our observations in larval brains of *cap-d3* dsRNA expressing larvae, the levels of NSPC death are significantly increased in the transheterozygotes, and this can be rescued by allowing larvae to develop on food containing AZT.

We also analyzed adult head sizes, comparing *cap-d3* transheterozygotes to heterozygotes and wild type adults that were reared on food containing DMSO or AZT (Fig. 7F). Similar to adult flies expressing *cap-d3* dsRNA, both the transheterozygotes and heterozygotes exhibited significantly decreased head sizes. However, development on food containing AZT partially rescued these defects.

We were unable to analyze the levels of retrotransposition in transheterozygous and heterozygous larval brains, using the gypsyCLEVR reporter, since all of the necessary alleles, including the *eyGMRGAL4* driver are on the second chromosome. Additionally, there are no additional *cap-d3* dsRNA lines available that efficiently knock down *cap-d3* (we have ordered and tested all available lines and found that only one dsRNA (VDRG stocks 29657 and 9402; two insertions of the same dsRNA) effectively targets *cap-d3*).

Comment 2) In Figs 2B and C, the effects on head size are shown for *smc2/+* or *cap-d2/+*. But in both of these panels, the "+" is a balancer (Cy0 or TM3). Thus formally, these effects on head size could be contributed by the balancer chromosome/*cap-d2*. This experiment should be redone without the presence of balancer (just cross the mutant/balancer to a wild type and take the non balancer heterozygous progeny and compare them to wild type controls).

RESPONSE: We have now examined head sizes in two additional lines, one expressing an additional *smc2* dsRNA, and one expressing an additional *cap-d2* dsRNA. Expression of the dsRNAs under the control of *eyGMRGAL4* results in pupal lethality. Therefore, we dissected

pupae from their pupal cases and now present pictures that show the severely decreased pupal head sizes (Fig. 2B, C, G). We also added qRT-PCR analyses of *smc2* or *cap-d2* transcripts in these lines, as compared to controls (Fig. 2D, H).

Comment 3) On page 3, top: 'statistically no different' is an odd phrasing. maybe not statistically significant or no difference or something like that.

RESPONSE: We have now changed this to read “not significantly different...”

Comment 4) Authors should acknowledge the caveat that some of the differences in phenotype between different Gal4 lines could be due to differences in expression level of the RNAi.

RESPONSE: We have now added the sentence “It is also possible that eyGAL4 and GMRGAL4 drive expression at higher levels in NSPCs than in other retrotransposition-permissible cells in the central brain.” On p. 25 of the discussion.

Comment 5) On page 20, typo noticed: "mobility in increased in" probably should read "mobility is increased in".

RESPONSE: We have now corrected this.

Comment 6) Finally, I very much appreciated the discussion section, which generally speaking is interesting and scholarly. The idea that CLEVR labeled larval neurons are lost normally during development, and that this could contribute to normal cell death is very interesting, and appropriate to speculate on. I do think though that the authors should discuss whether or not the effects of RTEs requires integration of de novo copies, just DNA damage from attempts at replication, or simply inflammatory effects from expression of dsRNAs or cDNAs. There is some literature that speaks to this, e.g. on LINE-1 elements driving cGAS/STING, on ERVs driving inflammation, and on ERVs causing DNA damage (e.g. in *Drosophila* neurodegeneration context).

RESPONSE: We thank the reviewer for their compliments regarding the discussion section. We have now expanded our discussion on the potential effects of RTE upregulation in the developing *Drosophila* brain and this can be found on p. 27.

Response to comments raised by Reviewer #2:

Comment 1) Should describe in more detail the expression patterns of ey-GAL4 and GMR-GAL4 in other tissues. For example, ey-GAL4 is expressed in the developing eye disc and phenotypes here could impact on the measurement taken of the adult heads.

RESPONSE: We have revised the section on p.16 that discusses these points: “Additionally, while *eyGAL4*, *GMRGAL4* and *optixGAL4* all drive expression in developing eye discs, *c253GAL4* does not⁴⁴, suggesting that *cap-d3 dsRNA* expression in eye discs is not required to cause microcephaly. It should be noted, however, that impaired development of eye discs can affect head development⁴⁵⁻⁴⁸, and it is therefore possible that *eyGAL4*, *GMRGAL4* and *optixGAL4*-driven expression of condensin dsRNAs in developing eye discs could contribute to the severity of the microcephaly observed.”

Comment 2) Scale bars in images should be included, especially when the zoom is obviously different within the same figure (e.g. Figure 4B).

RESPONSE: We have now added scale bars to every image taken with the confocal microscope. In response to the reviewer’s concern that some of the images (Fig 4B) appear to have zoomed in images, this is actually due to how the experiment is performed. Following the immunostaining protocol, larval brains are placed on the slide, in between the slide and the coverslip. To prevent squashing, keep the coverslip in place, and allow for full imaging of the whole brain, we place vacuum grease and small shards of broken glass at each corner of the coverslip. However, some brains will become slightly more compressed than others, depending on their position on the slide. This causes them to look slightly bigger when the Maximal projection is produced.

Comment 3) Is cell death occurring in ganglion mother cells (GMCs) but not immature neurons? This would be good to clarify. Could be tested by staining for DCP1, Pros and Elav.

RESPONSE: We thank the reviewer for this question and agree that understanding the specific cell types that are undergoing cell death is an important characterization of this microcephaly model, however one that we cannot comprehensively answer at this time. We performed immunofluorescence analyses to detect *dcp1*, *ElaV*, and *Pros* OR *dcp1*, *Dpn*, and *Pros*, but we were not able to obtain the resolution necessary to make any conclusions on the cell types expressing *dcp1* (attached at the end of this responses document). As demonstrated in the attached figures, the common molecular signatures used to identify Type I and II neuroblasts, ganglion mother cells, and immature neural precursors overlap in many ways that can make it hard to differentiate between them. Further, newly identified markers that may assist in this differentiation do not have commercially available antibodies. We hope to be able to fully investigate the cell types in the future, as we obtain non-commercial antibodies and reporter stocks, optimize our techniques, and introduce new techniques (such as single cell RNA-seq), in future work.

Comment 4) It is surprising that RTE elements are active in the control GFP dsRNA experiments. Is general activation of RNAi machinery resulting in RTE expression? Testing whether this happens with a different RNAi control and in other developing tissues would be helpful for this study and more generally for RNAi studies. Related to this (Figure 7) - What about *Dpn+* *Dcp1+* numbers in brains not expressing any RNAi? Is generic RNAi also increasing apoptosis?

RESPONSE: We have added experiments involving a no RNAi control (*eyGMR*) and analyzed the numbers of *dcp1+* NSPCs and *dcp1+* NSPC-adjacent cells in larval central brains, comparing results to brains expressing control *GFP dsRNA* and brains expressing *cap-d3 dsRNA* (Fig. 5E and 5F). Additionally, we analyzed numbers of NSPCs that were positive for a retrotransposition

event in larval brains expressing *UAS-gypsyCLEVR*, in the absence of RNAi, and compared results to larval brains expressing *UAS-gypsyCLEVR* in combination with *GFP* or *cap-d3 dsRNA* (Fig. 6C). The results demonstrated that retrotransposition events occur at similar levels in brains that do not express any dsRNA and those expressing control *GFP dsRNA*. Likewise, levels of cell death in NSPCs or their neighbors were similar between brains that do not express any dsRNA and those expressing control *GFP dsRNA*. Therefore, these results suggest that RNAi expression alone does not cause increased RTE activity or concomitant NSPC cell death, and that these effects are unique to *cap-d3 dsRNA* expression.

Given that the focus of this manuscript is on the developing brain, we did not perform experiments in other developing tissues, and we hope to perform these studies in future experiments.

Comment 5) Discussion – ‘driven by eyGAL4, GMRGAL4, results in significant decreases in adult brain volumes and head sizes, without affecting wing size (Fig. 1-3).’ Isn’t this just because GAL4 isn’t expressed there? What point is being made here?

RESPONSE: We agree with the reviewer that this was confusing, and we have now removed these sentences from the manuscript, and removed the data from the supplementary figures.

Comment 6) ‘The fact that eyGAL4, GMRGAL4 drives in a large area of the larval central brain (Fig. 4A), but retrotransposition events were only observed in NSPCs, suggests that these cell types may be the only cell types that permit RTE mobilization in the developing brain.’ – could this not just be due to differences in the level of GAL4 expression?

RESPONSE: We have now added the sentence “It is also possible that eyGAL4 and GMRGAL4 drive expression at higher levels in NSPCs than in other retrotransposition-permissible cells in the central brain.” On p.25 of the discussion.

Comment 7) ‘Recent evidence demonstrated that knockdown of condensin I subunit Cap-g in postmitotic neurons affected the expression of a subset of neuronal genes.’ – It also results in the ectopic expression of non-neuronal genes – could this be linked to RTE expression?

RESPONSE: We have now changed this sentence to read “Recent evidence demonstrated that knockdown of condensin I subunit Cap-g in post-mitotic neurons affected the expression of both neuronal and non-neuronal genes.” on p. 26. Following this statement, we have discussed the possibility that retrotransposition could regulate mosaic gene expression in larval neurons, but that these effects would most likely be transient, given our data showing that increased retrotransposition leads to NSPC death.

Response to comments raised by Reviewer #3:

Comment 1) The first main conclusion of the paper, as outlined in the first paragraph of the discussion, is that tissue-specific knockdown of condensin I and II in the brain causes microcephaly. However, it is noted elsewhere in the paper that ubiquitous loss of condensin function causes a general reduction in organism size, and that this reduction is more prominent in male flies. Unless there is some evidence for brain-specific loss of condensin activity occurring

naturally (in *Drosophila* or any organism), the purported link between condensin loss and microcephaly is entirely artificial.

RESPONSE: The reviewer is correct, in that we have previously published that homozygous *cap-d3* mutants exhibited decreased body weights, suggesting they were smaller in size. However, we wrote in the discussion on p. 23 that “Interestingly, germline condensin mutations identified in microcephaly patients caused some, but not all, of the patients to exhibit significantly decreased stature, or even dwarfism¹². It is perplexing that some germline mutations that cause decreased expression of condensin proteins, which function in every cell type, cause reductions in brain volume in primary microcephaly patients without affecting overall body size.” We then go on to explain our hypothesis that “RTE expression and activity may be higher in developing human brain tissues than in other somatic tissues within the body, and thus, the depletion of condensin protein increases RTE expression and activity to very high levels that cannot be tolerated. This may not happen in other organs where baseline RTE expression/activity levels are much lower, such that condensin insufficiency still results in higher RTE expression/activity, but it remains below a threshold that can still be tolerated by the cells. It is also possible that in some organisms, the mechanisms that regulate condensin expression/ activity may become dysregulated specifically in the brain.”

Comment 2) The second major conclusion of the paper is that TE expression and activity downstream of condensin loss “leads to” increased cell death. This claim implies a causal relationship between TE expression/activity and apoptosis, but no evidence is supplied to support causation. Indeed, *Ago2* mutant flies are used as a model of TE dysregulation in this manuscript. Do these flies exhibit increased apoptosis in the brain and decreased head sizes? If not, it is much more likely that cell death and abnormal brain development lie downstream of another function of condensin, aside from its role in TE suppression. However, there is no exploration of any other role for condensins in preventing apoptosis.

RESPONSE: We have addressed this comment with the following experiments:

- 1) In our first submission, we showed in Figure 7A and 7B that allowing larvae to develop on food containing AZT abrogates that increased levels of cell death in NSPCs and NSPC-adjacent cells observed in *cap-d3 dsRNA* expressing larval brains. This suggests that RTE retrotransposition is necessary for the observed increases in cell death. We have now added experiments involving *cap-d3* transheterozygous mutant larvae (Fig. 7C) which show very similar results.
- 2) We now show in Supplemental Figure 5B that NSPC death is also increased in *ago2* transheterozygous mutant larval brains, and allowing these larvae to develop on food containing AZT almost completely rescues the observed increases in cell death.
- 3) We now show in Supplemental Figure 5C that adults which are transheterozygous or heterozygous for a small deletion (454) and/or a large deletion (Df (3L) BSC558) exhibit smaller head sizes when compared to wild type flies. Furthermore, the phenotypes are rescued when flies are reared on food containing AZT.

Comment 3) The third major conclusion is that NRTI ingestion during development inhibits TE retrotransposition as well as TE expression, and that this re-establishment of TE control results in normal head sizes in adult flies. This claim is highly problematic. NRTIs are reverse

transcriptase inhibitors, and have not, to my knowledge, been demonstrated previously to inhibit TE expression. While NRTIs at very high doses can inhibit DNA polymerases, they are not characterized as RNA polymerase inhibitors. There is no explanation given by the authors as to why NRTIs might specifically repress TE expression, but not the expression of cellular genes (eg Ago2, which is shown in Sup Fig 6 to be unperturbed by AZT treatment). If NRTI treatment does, by some unknown mechanism, directly and specifically inhibit TE expression in addition to reverse transcription, it would be extremely interesting to the TE field. NRTIs are often used to support claims that retrotransposition or the generation of TE cDNAs has some effect on cellular function or disease state, and the demonstration that they actually impact TE expression would change the interpretation of numerous previous results. Unfortunately the claimed reduction in TE expression by NRTIs is only demonstrated explored in the intestines of ago2 mutant flies by qPCR, and is not demonstrated in condensin deficient brains exhibiting apparent rescue of head size by NRTIs. It therefore is equally likely that head size rescue is a side-effect of NRTI administration unrelated to TE expression or retrotransposition.

RESPONSE:

1) We have previously published that treatment of human colon adenocarcinoma cells with the NRTIs, zidovudine (AZT) and didanosine does, in fact, decrease transcript levels of LINE-1 (Ward et al. Nuc Acids Res 2022). However, we have not previously shown that this occurs in *Drosophila*. While our preliminary data suggest that rearing ago2 mutants on food containing AZT results in decreased transcripts of several RTE families, we have only performed these experiments using heterozygotes. Therefore, we have decided to remove the data from the supplementary figures, and we plan to explore these findings in a more comprehensive manner in future experiments using several mutants/ RNAis of multiple RTE regulators.

2) We have now added experiments to Figure 6D to demonstrate that NRTI addition to the food represses the increased levels of retrotransposition observed in larval NSPCs of Cap-d3 insufficient brains, returning them to levels observed in *GFP dsRNA* expressing controls. Interestingly, these experiments also revealed that AZT does not completely abrogate retrotransposition events in developing control or *cap-d3* insufficient brains. The most likely explanation for this result is that the retrotransposition observed in Cap-d3 insufficient brains occurs after the development of the first instar larval mouth hooks, allowing larvae time to ingest NRTI which then inhibits future retrotransposition events. Remaining retrotransposition events observed using the gypsyCLEVR reporter would have occurred prior to this developmental stage, thus causing them to be unaffected by the addition of NRTIs to the food. These points are discussed on p. 19 and p. 25, and we thank the reviewer for helping us to uncover this fascinating aspect of the project.

eyGAL4, GMRGAL4 >
UAS-mCherry-dsRNA

eyGAL4, GMRGAL4 >
UAS-cap-d3-dsRNA

eyGAL4, GMRGAL4 >
UAS-mCherry-dsRNA

eyGAL4, GMRGAL4 >
UAS-cap-d3-dsRNA

[schematic redacted]

REVIEWER COMMENTS

Reviewer #1 (Remarks to the Author):

The authors have addressed many of the concerns that I raised in the previous review except for one, which was a major issue in my view. The fundamental core of this manuscript is examination of a series of phenotypes, and biological consequences of mutations in this condensin gene. The entire manuscript, and all of its conclusions, stem from this fundamental claim. The evidence for this comes from examination of an RNAi line that targets the gene, and from mutant alleles that target the gene. In principle, this convergence would be sufficient, but in my previous review, I pointed out that the RNAi approach rested on a single RNAi line. It is well known that RNAi approaches can have off target effects, and therefore it is standard to use more than one RNAi line. The authors were unable to do this because there are no additional extant RNAi lines that are effective. The other approach that I suggested is to use more than one allele and then do a full complementation test, or do a rescue by supplying a rescuing transgene. The challenge with complementation is that the Df and the hypomorph are each at least semi dominant. So while the authors did test both alleles as heterozygotes, and tested the trans-heterozygous combination (Fig 7), the dominance of both alleles prevents conclusions about failure to complement. I appreciate that the experiments are challenging, but nevertheless I remain concerned that there is insufficient genetic evidence presented here to fully support this key set of conclusions.

There are other options available to the authors. While Df/hypomorph is no more severe than each of the dominant alleles in heterozygous state, the authors could test (as I suggested) whether the Df/hypomorph precludes further impact of the RNAi line. If Df/hypomorph is no more severe than Df/+ or hypomorph/+, then further KO by RNAi should have no effect. Or the authors could generate additional genetic reagents to bolster the genetic evidence.

I appreciate that it can be frustrating when the genetic tools are not sufficiently precise to answer the question, but I think the evidence needs to be more secure to support the fundamental conclusions made here.

Reviewer #2 (Remarks to the Author):

Crawford and colleagues have significantly revised their manuscript describing their work investigating the role of condensin in silencing retrotransposable elements (RTEs) in the developing brain of *Drosophila*. They have addressed all of my specific comments to a sufficient degree that I would support publication. It is a pity that they did not manage to identify whether immature neurons were also undergoing cell death. However, this is not not an essential aspect of this study.

Reviewer #3 (Remarks to the Author):

The authors have made a substantial effort to address my concerns.

For point 1, it may be helpful to note in the introduction of the paper that in human patients, the brain seems more sensitive to condensin mutation than other tissues of the body, evidenced by condensin mutations frequently causing microcephaly but less frequently causing an over-all reduction in size. This would help justify the brain-specific knockout in the fly model.

For point 2, I appreciate the additional experimental work. The results taking advantage of Ago2 mutant flies as an independent model of TE dysregulation greatly strengthen the paper.

For point 3, I understand the decision to remove the data on RT inhibitors decreasing TE expression, and look forward to follow-up work on this phenomenon. The insight that larval development dictates the ability to ingest AZT and therefore impacts the timing of detected retrotransposition events is very interesting and I enjoyed the explanation of this result in the paper.

Response to Reviewer Comments

We thank all of the reviewers for their time and assistance in helping us to improve our manuscript. We have addressed all comments below:

Response to comments raised by Reviewer #1:

Comment 1) The authors have addressed many of the concerns that I raised in the previous review except for one, which was a major issue in my view. The fundamental core of this manuscript is examination of a series of phenotypes, and biological consequences of mutations in this condensin gene. The entire manuscript, and all of its conclusions, stem from this fundamental claim. The evidence for this comes from examination of an RNAi line that targets the gene, and from mutant alleles that target the gene. In principle, this convergence would be sufficient, but in my previous review, I pointed out that the RNAi approach rested on a single RNAi line. It is well known that RNAi approaches can have off target effects, and therefore it is standard to use more than one RNAi line. The authors were unable to do this because there are no additional extant RNAi lines that are effective. The other approach that I suggested is to use more than one allele and then do a full complementation test, or do a rescue by supplying a rescuing transgene. The challenge with complementation is that the Df and the hypomorph are each at least semi dominant. So while the authors did test both alleles as heterozygotes, and tested the trans-heterozygous combination (Fig 7), the dominance of both alleles prevents conclusions about failure to complement. I appreciate that the experiments are challenging, but nevertheless I remain concerned that there is insufficient genetic evidence presented here to fully support this key set of conclusions.

There are other options available to the authors. While Df/hypomorph is no more severe than each of the dominant alleles in heterozygous state, the authors could test (as I suggested) whether the Df/hypomorph precludes further impact of the RNAi line. If Df/hypomorph is no more severe than Df/+ or hypomorph/+, then further KO by RNAi should have no effect. Or the authors could generate additional genetic reagents to bolster the genetic evidence.

RESPONSE: We would like to point out that we did provide rescue experiments using a GFP-Cap-D3 allele, and this data is presented in Figure 1G; we apologize for not pointing this out in the first response to reviewer comments. These results suggest that the Cap-D3 RNAi is specific, and knockdown of Cap-D3 is responsible for causing the observed phenotypes, since co-expression of UAS-GFP-Cap-D3 rescues the microcephaly phenotype. Furthermore, we have presented evidence that knockdown of other condensin subunits also result in the same phenotypes. Therefore, the likelihood of off-target effects of 1 RNAi (which we can rescue with a UAS-Cap-D3 transgene) being responsible for causing the same phenotypes as knockdown of 2 additional condensin proteins (using 2 other RNAis per subunit), is extremely small.

Additionally, we have performed experiments using two different mutant alleles, in the heterozygous and/or transheterozygous state. The reviewer is correct in that the data suggest that these alleles are each dominant. However, the transheterozygotes and heterozygotes all

exhibit microcephaly phenotypes and this is rescued by treatment with AZT (Figure 7C and Figure 7F). This provides additional genetic evidence that knockdown of Cap-D3 (whether using heterozygotes or transheterozygotes) results in the same phenotypes that we see following RNAi expression.

Response to comments raised by Reviewer #2:

Comment 1) Crawford and colleagues have significantly revised their manuscript describing their work investigating the role of condensin in silencing retrotransposable elements (RTEs) in the developing brain of *Drosophila*. They have addressed all of my specific comments to a sufficient degree that I would support publication. It is a pity that they did not manage to identify whether immature neurons were also undergoing cell death. However, this is not not an essential aspect of this study.

RESPONSE: We thank the reviewer for their comments and as more tools become available (either engineered in our lab or donated by other labs), we will continue to perform experiments to identify the exact cell types that are dying in the condensin insufficient brains.

Response to comments raised by Reviewer #3:

The authors have made a substantial effort to address my concerns.

For point 1, it may be helpful to note in the introduction of the paper that in human patients, the brain seems more sensitive to condensin mutation than other tissues of the body, evidenced by condensin mutations frequently causing microcephaly but less frequently causing an over-all reduction in size. This would help justify the brain-specific knockout in the fly model.

For point 2, I appreciate the additional experimental work. The results taking advantage of Ago2 mutant flies as an independent model of TE dysregulation greatly strengthen the paper.

For point 3, I understand the decision to remove the data on RT inhibitors decreasing TE expression, and look forward to follow-up work on this phenomenon. The insight that larval development dictates the ability to ingest AZT and therefore impacts the timing of detected retrotransposition events is very interesting and I enjoyed the explanation of this result in the paper.

RESPONSE: We thank the reviewer for helping us to clarify this important point. On page 3 of the introduction, we have now added, “Interestingly, the identified mutations/deletions caused microcephaly in all patients, but did not affect overall body size in all of the patients, suggesting that the brain may be more sensitive to condensin mutation than other tissues.”

REVIEWERS' COMMENTS

Reviewer #1 (Remarks to the Author):

The author's response satisfies my critique totally. Although the authors noted in their most recent response that they forgot to note the addition of the gfp tagged rescue in their earlier response to reviewers, it was my error to miss that in the revised manuscript. I have no further substantive critiques of this manuscript. Its a nice story.